# Unified Latent Steering and Residual Refinement for Online Improvement of Diffusion Policy Models

## Abstract

Imitation learning has driven major advances in robotic manipulation by exploiting large and diverse demonstrations, yet policies trained purely by imitation remain brittle under distribution shift and novel scenarios, making online improvement essential. Directly finetuning the parameters of modern large policies is prohibitively sample inefficient and computationally expensive, while recent finetuning-free adaptation methods either fail to exploit the multimodal distributions learned by pretrained policies or remain confined to the coverage of demonstrations. We propose **USR**, a **U**nified framework for latent **S**teering and residual **R**efinement that enables efficient online improvement of diffusion policy models. A lightweight actor jointly outputs latent noise to steer the diffusion process toward promising modes and residual corrections to adapt beyond the diffusion policy's support, combining stable mode selection with flexible refinement. This unified design stabilizes training and fully leverages both components. Experiments on standard benchmarks and our MultiModalBench demonstrate USR's state-of-the-art performance. Furthermore, we validate its real-world applicability by improving a Vision-Language-Action (VLA) model on a physical robot, setting a new paradigm for sample-efficient adaptation of diffusion-based policies.

## 1 Introduction

A longstanding ambition in robotics is to endow machines with human-like manipulation across diverse environments (Billard & Kragic, 2019). Recent progress in imitation learning, fueled by advances in architectures (Vaswani et al., 2017; Ho et al., 2020) and large-scale demonstrations (O'Neill et al., 2024; Khazatsky et al., 2024), has enabled policies capable of dexterous hand control (Arunachalam et al., 2022), household visuomotor skills (Fu et al., 2024), and even emerging generalist abilities (Black et al., 2024; Bjorck et al., 2025; Cheang et al., 2025). Despite these advances, progress has been mostly demonstrated in controlled settings, while open-world manipulation presents a much broader long-tail of objects, layouts, contacts, and partial observability (Zitkovich et al., 2023). Unlike humans who adapt within a few interactions, imitation-learned policies are fixed once training ends. Their behavior distributions are anchored to the demonstrations, making unseen situations hard to handle. Although there are offline-to-online RL methods designed for similar settings (Nakamoto et al., 2023; Zhou et al., 2024), applying them to large policy models requires updating parameter-heavy networks. Even with carefully designed fine-tuning techniques (Hu et al., 2022), such updates incur significant computational overhead and large sample demands (Wagenmaker et al., 2025). These constraints motivate alternatives that can deliver rapid behavioral adaptation without finetuning the large policy model.

Existing finetuning-free policy adaptation methods broadly fall into two categories. The first steers the base policy's sampling process, exploiting the multimodality[1] of large policy models by biasing sampling toward promising modes (Nakamoto et al., 2024; Wagenmaker et al., 2025; Du & Song, 2025). The second adds a residual actor that refines the output of the frozen base policy, adjusting

---

[1] In this paper, we use the terms *multimodal* and *multimodality* to mean action distributions with multiple behavior modes. This is distinct from the common usage of *multimodal* to describe models that integrate different input modalities such as vision and language.

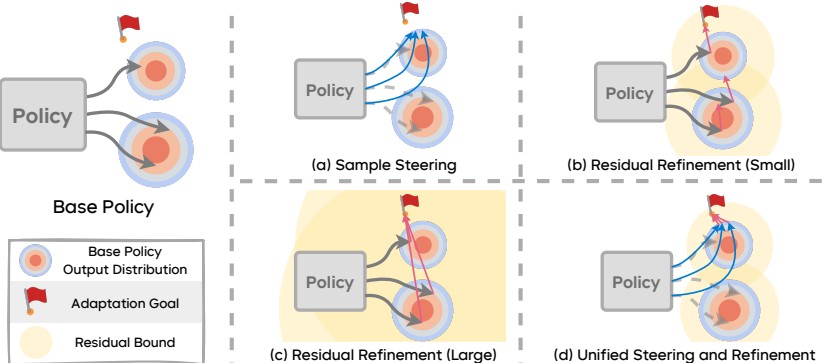

Figure 1: **Comparison of finetuning-free online adaptation methods.** The base policy has two modes, while the optimal region lies outside the upper mode. (a) Sample steering stays within the base support. (b) Residual refinement with small bounds cannot cross the gap. (c) Large bounds cross the gap but explore inefficiently. (d) USR combines sample steering and residual refinement for stable, sample-efficient adaptation.

actions towards more favorable directions (Johannink et al., 2019; Yuan et al., 2024). In practice, residual methods often constrain the adjustment with a bound to promote efficient exploration. While effective, both approaches face limitations. In Figure 1, we show a single-step decision problem where the base policy has two action modes, and the goal for adaptation lies outside one of them. Sample steering can bias sampling towards the nearest mode but remains confined to the base policy's support. Residual refinement with small bounds cannot cross the mode boundary, whereas setting large bounds permit crossing but induce inefficient exploration. These limitations highlight the need for an approach that can balance stable mode selection with flexible refinement beyond the pretrained distribution.

To address the limitations of existing finetuning-free online adaptation methods, we propose **USR**, a **U**nified framework for latent noise **S**teering and residual **R**efinement. USR augments a pretrained diffusion policy model with a single lightweight actor that jointly produces initial noise to steer the diffusion process and residual corrections to refine its outputs. The noise output allows the policy to exploit the multimodal structure of diffusion policy models, guiding trajectories toward promising modes, while the residual component provides the flexibility to adapt beyond the support of the base policy when necessary. This unified formulation combines the strengths of both perspectives, mode selection and action refinement, within a stable reinforcement learning framework. As a result, USR enables pretrained policies to rapidly adjust to novel environments, improving task success with only a modest number of interactions and without modifying the underlying large policy model.

We validate USR through experiments on three benchmarks: our proposed MultiModalBench, the Adroit suite (Rajeswaran et al., 2017) of dexterous hand tasks, and two tasks from ManiSkill (Gu et al., 2023; Mu et al., 2021; Tao et al., 2024). MultiModalBench highlights the challenge of selecting among multiple demonstration modes, Adroit tests adaptation under human-provided demonstrations, and ManiSkill covers settings with mostly single-modal data. We also extend our evaluation to the physical world, demonstrating that USR effectively improves a pre-trained VLA model on a real robot. Across all settings and under both state and visual observations, USR achieves consistently higher success and superior sample efficiency compared to prior methods. Qualitative analysis shows that latent steering reliably selects the correct behavioral mode while residual refinement makes fine-grained corrections beyond the base policy's support.

Our contributions are fourfold:

- We identified complementary limitations of online adaptation methods in manipulation: sample steering is constrained by the base policy, and residual refinement requires fragile step-size tuning.
- We proposed **USR**, a unified online adaptation algorithm for diffusion policies that employs a single lightweight actor to jointly generate noise and refine trajectories, enabling multimodal steering and controlled policy deviation without parameter updates of the pretrained policy.
- We released **MultiModalBench**, a benchmark of six robot manipulation tasks with multiple demonstration modes, providing the first systematic testbed for multimodal policy adaptation.

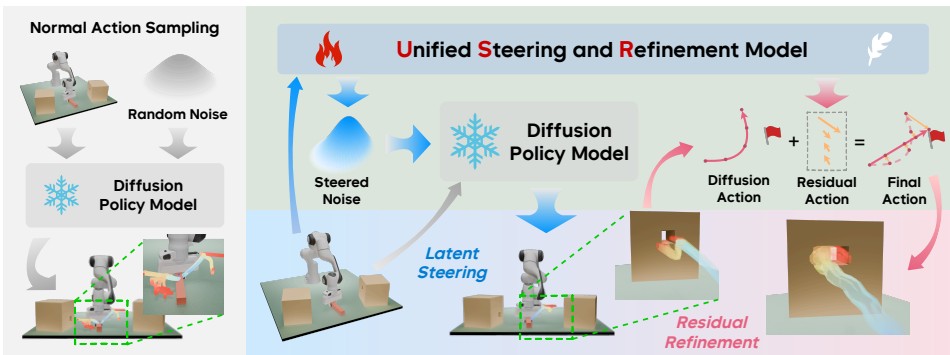

Figure 2: **How USR improves a bimodal base policy on the PegInsertionSideStrict task from MultiModalBench.** The pretrained diffusion policy model has two modes: inserting a peg into the left box or the right box. Latent steering first steers the policy to consistently select the correct (right) mode, but the trajectories remain imprecise. Residual refinement then applies fine-grained corrections, guiding all trajectories to the target hole. This representative scene shows how USR combines coarse sample steering with precise action refinement for task success.

- We demonstrated the real-world applicability of USR by effectively improving a VLA model on a physical robot, validating its potential for scalable fine-tuning of behavioral foundation models.

## 2 RELATED WORK

**Policy improvement with reinforcement learning.** Reinforcement learning is widely used to adapt pre-trained policies. Existing methods can be broadly divided into two categories based on whether they finetune the base policy. The first category directly finetunes pretrained imitation learning (Ren et al., 2024; Chandra et al., 2025), offline reinforcement learning (Nakamoto et al., 2023; Zhou et al., 2024), or Vision-Language-Action models (SimpleVLA-RL Team, 2025; Lu et al., 2025; Mark et al., 2024) using online RL gradients. The second category improves policy performance without modifying the base policy itself, often by learning a residual (Yuan et al., 2024; Ankile et al., 2024) or auxiliary policy (Wagenmaker et al., 2025) on top of the fixed base. Our method belongs to this second category and introduces a novel decomposition of policy improvement into latent steering and action refinement, enabling comprehensive and efficient enhancement of the base policy's performance.

**Noise optimization in generative models.** Steering and improving generative models via noise-space optimization has been widely studied across domains. In image synthesis, recent work (Eyring et al., 2024; Mao et al., 2024; Samuel et al., 2024) shows that optimizing the initial diffusion noise to maximize downstream image-quality metrics can yield substantial gains. In robotics and control, Singh et al. (2020) trains a normalizing-flow policy on offline data and then runs reinforcement learning directly in the policy's noise space to improve online performance. Most closely related to our setting, DSRL (Wagenmaker et al., 2025) optimizes the noise for Diffusion Policy via RL to enhance control outcomes. We identify a key limitation of DSRL: by optimizing noise while keeping the base policy fixed, it is highly constrained by the support of the base policy's action distribution, which caps performance at the quality of the imitation demonstrations. We address this by introducing a unified framework that combines latent steering with explicit action refinement, enabling elegant exploration and yielding stronger, more sample-efficient online improvements.

## 3 PROBLEM FORMULATION

We consider a discounted Markov Decision Process (MDP) $\mathcal{M}(\mathcal{S}, \mathcal{O}, \mathcal{A}, p_0, P, r, \gamma)$. At time $t$, the environment is in state $s_t \in \mathcal{S}$, while $s_t \sim p_0$ (if $t = 0$) or $s_t \sim P(\cdot|s_{t-1}, a_{t-1})$, the agent receives observation $o_t \in \mathcal{O}$, choose action $a_t \sim \mathcal{A}$, and transitions to $s_{t+1}$. In our setting, the agent is equipped with a pretrained diffusion policy model $\pi_{dp}$ obtained through imitation learning on offline demonstrations. While $\pi_{dp}$ captures diverse behavior from demonstrations, it may fail to achieve the goal in the current environment due to distribution shift or incomplete coverage of pretrained

behaviors. The objective of online adaptation is therefore to enhance $\pi_{\text{dp}}$ using online interactions so that the resulting policy $\pi_{\text{new}}$ maximizes the expected discounted return:

$$J(\pi_{\text{new}}) = \mathbb{E}_{s_0 \sim p_0, \pi_{\text{new}}, P} \left[ \sum_{t=0}^{\infty} \gamma^t r(s_t, a_t) \right] . \tag{1}$$

# 4 UNIFIED STEERING AND REFINEMENT FRAMEWORK

To address the complementary limitations of sample steering and residual refinement, we introduce USR, a unified framework for online adaptation of pretrained diffusion policy models. USR employs a lightweight actor that jointly outputs latent noise to steer the diffusion sampling process and a residual correction to further refine the resulting action. This unified formulation leverages the complementary strengths of both noise-space steering and residual refinement, while avoiding their respective limitations. We begin with the unified actor design in Section 4.1, then describe the combined critic design and critic learning in Section 4.2 and the actor learning procedure in Section 4.3. Pseudocode of the complete algorithm is provided in Appendix B.

## 4.1 UNIFIED LATENT STEERING AND RESIDUAL REFINEMENT

At the core of USR is a single, lightweight actor, $\pi_\theta(o_t)$, that takes the current observation $o_t$ and outputs a combined action $a_t^{\text{comb}}$, which is a concatenation of two components, a latent noise $w_t \in \mathcal{W}$ and a residual action $a_t^{\text{res}} \in \mathcal{A}$:

$$a_t^{\text{comb}} = [w_t, a_t^{\text{res}}] \sim \pi_\theta(\cdot | o_t) . \tag{2}$$

The latent noise $w_t$ is constrained within a bounded space $[-b_w, b_w]$. These two components are then used to adjust the base diffusion policy $\pi_{\text{dp}}$ in a two-stage process:

**Latent Steering:** The latent noise vector $w_t$ is used as the initial noise to start the denoising process of $\pi_{\text{dp}}$. This steers the base policy to generate a biased action $\tilde{a}_t$:

$$\tilde{a}_t = \pi_{\text{dp}}(o_t, w_t) . \tag{3}$$

By replacing standard Gaussian noise with learned noise, we bias sampling toward promising modes rather than relying on the base policy to stochastically land in one of them.

**Residual Refinement:** The residual action vector $a_t^{\text{res}}$ is then added to the steered action $\tilde{a}_t$ to make fine-grained corrections. A residual scale $\alpha$ controls the magnitude of this adjustment. The final action $a_t$ executed in the environment is

$$a_t = \tilde{a}_t + \alpha \cdot a_t^{\text{res}} . \tag{4}$$

To ensure stable learning, especially at the beginning of training when the residual output is randomly initialized, we adopt the *progressive exploration* strategy from Policy Decorator (Yuan et al., 2024). Instead of always applying the residual refinement, we introduce it gradually. During online rollouts for training, the residual action $a_t^{\text{res}}$ is added with a probability $\epsilon$ that increases linearly from 0 to 1 over a set number of environment steps, $H$. This allows the agent to initially rely on the more stable base policy and avoid early failures, ensuring it continues to receive success signals. The final behavioral action $a_t$ is therefore determined as:

$$a_t = \begin{cases} \pi_{\text{dp}}(o_t, w_t) + \alpha \cdot a_t^{\text{res}} & \text{Uniform}(0, 1) < \epsilon \\ \pi_{\text{dp}}(o_t, w_t) . \end{cases} \tag{5}$$

Together, this unified framework allows USR to first make a coarse selection among the diverse behaviors learned by the base policy via steering, and then apply a fine-grained correction that can even push the final action beyond the original support of $\pi_{\text{dp}}$.

## 4.2 CRITIC LEARNING MECHANISM

A key challenge in learning the unified actor is providing a stable and efficient gradient signal. Backpropagating through the iterative denoising process of $\pi_{\text{dp}}$ is computationally expensive and

often numerically unstable. While a standard actor-critic algorithm in the latent space bypasses this issue, it is highly sample-inefficient because it must redundantly explore different noise vectors that map to similar actions. To circumvent this, USR employs a two-critic architecture inspired by the noise-aliased distillation in DSRL (Wagenmaker et al., 2025), adapted for our unified framework.

**Environment Critic** $Q_\phi^{\mathcal{A}}(o, a)$**:** This critic operates in the environment's action space $\mathcal{A}$. Its purpose is to learn the value of the final, executed actions $a_t$. It is trained using standard off-policy temporal difference (TD) learning from transitions $(o_t, a_t, r_t, o_{t+1})$ stored in a replay buffer $\mathcal{D}$. The loss for the environment critic is:

$$\mathcal{L}_{\text{TD}}(\phi) = \mathbb{E}_{(o_t, a_t, r_t, o_{t+1}) \sim \mathcal{D}} \left[ \left( Q_\phi^{\mathcal{A}}(o_t, a_t) - y_t \right)^2 \right] , \qquad (6)$$

where the TD target $y_t$ is computed as $y_t = r_t + \gamma(1 - d_t)Q_{\phi'_{\text{target}}}^{\mathcal{A}}(o_{t+1}, a'_{t+1})$, with $a'_{t+1}$ being the next action from the actor policy and $d_t$ being the episode termination signal.

**Combined Critic** $Q_\psi^{\mathcal{C}}(o, a^{\mathbf{comb}})$**:** This critic operates directly in the actor's output space, evaluating the combined action $a_t^{\text{comb}} = [w_t, a_t^{\text{res}}]$. Instead of learning from sparse rewards via TD learning, it is trained to distill the value from the environment critic. This provides a direct and sample-efficient gradient path to the actor. The distillation loss is formulated as

$$\mathcal{L}_{\text{distill}}(\psi) = \mathbb{E}_{o \sim \mathcal{D}, a^{\text{comb}} \sim U} \left[ \left( Q_\psi^{\mathcal{C}}(o, a^{\text{comb}}) - Q_\phi^{\mathcal{A}}(o, a_{\text{env}}) \right)^2 \right] , \qquad (7)$$

where $a_{\text{env}} = \pi_{\text{dp}}(o, w) + \alpha \cdot a^{\text{res}}$ is the final action computed from a randomly sampled combined action $a^{\text{comb}} = [w, a^{\text{res}}]$, and the environment critic $Q_\phi^{\mathcal{A}}$ is held fixed during the distillation update. The combined critic updates are applied $N_D$ times per iteration. This dual-critic setup decouples the complex dynamics of the diffusion policy from the actor's learning process, enabling stable and efficient training.

### 4.3 ACTOR LEARNING

With the combined critic $Q_\psi^{\mathcal{C}}$ providing a value estimate for any combined action, the actor $\pi_\theta$ can be trained to maximize the expected return using policy gradients. We adopt the Soft Actor-Critic (SAC) (Haarnoja et al., 2018) objective to encourage exploration through entropy maximization. The actor's objective is to maximize:

$$J(\theta) = \mathbb{E}_{o_t \sim \mathcal{D}, a_t^{\text{comb}} \sim \pi_\theta} \left[ Q_\psi^{\mathcal{C}}(o_t, a_t^{\text{comb}}) + \beta \mathcal{H}(\pi_\theta(\cdot|o_t)) \right] \qquad (8)$$

where $\mathcal{H}$ is the policy's entropy and $\beta$ is a temperature parameter that can be automatically tuned. The gradient flows directly from the combined critic to the actor, bypassing the diffusion policy entirely and allowing for efficient updates to the lightweight actor network.

## 5 EXPERIMENTS

Our experiments are designed to empirically answer the following questions: 1) Can our method USR effectively improve pre-trained diffusion policy models on the simulation tasks of **Multi-ModalBench**, **AdroitHand**, and **ManiSkill** under both state and visual observations? (Section 5.2) 2) How do key hyperparameters influence the performance of USR? (Section 5.3) 3) How does USR improve the performance of pre-trained diffusion policy models? (Section 5.4) 4) Can USR be applied to real-world manipulation and improve Vision-Language-Action (VLA) models? (Section 5.5)

### 5.1 SIMULATION EXPERIMENTS SETUP

#### 5.1.1 TASK DESCRIPTION

Our experiments are conducted on 6 simulation tasks from **MultiModalBench**, 3 simulation tasks from **Adroit** (Rajeswaran et al., 2017), and 2 simulation tasks from **ManiSkill** (Gu et al., 2023). Refer to Figure 3 for task visualizations.

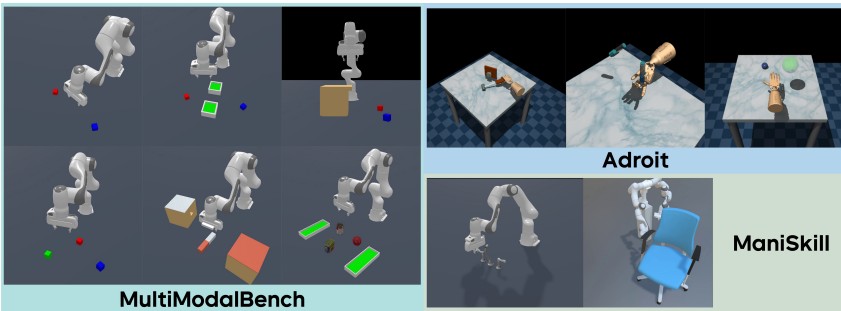

Figure 3: Illustration of the tasks used in our experiment, including six **MultiModalBench** tasks, three **AdroitHand** tasks, and two **ManiSkill** tasks. For each MultiModalBench task, visualizations of all behavior modes and the adaptation target mode (*Strict*) are provided in Appendix Figure 9.

**MultiModalBench**. We build six simulation tasks on top of SAPIEN (Xiang et al., 2020) to form MultiModalBench, including *PickCube*, *StackThreeCube*, *PlaceTwoCube*, *PegInsertionSide*, *OpenBoxPlaceCube*, and *SortYCB*, each containing multiple behavior modes. For each task, we collect expert datasets using an off-the-shelf motion planner, which include successful trajectories for all behavior modes. Tasks with the *Strict* suffix indicate that, among all behavior modes, only a single mode is considered successful. Visualizations of all modes including the adaptation target mode for each task are provided in Appendix Figure 9. The adaptation objective is to maximize the success rate on these *Strict* tasks, evaluating the ability to both steer multimodal policies toward the desired behavioral mode and refine actions to surpass the base policy's performance. We use *sparse reward* for all of our experiments.

**AdroitHand**. We evaluate on three AdroitHand simulation tasks, *Pen*, *Hammer*, and *Relocate*, which require solving dexterous manipulation with a 24-DoF hand simulator. Following the setup of Rajeswaran et al. (2017), we use 25 human demonstrations for training the base policy. We exclude the *Door* task since the base policy already achieves near-perfect performance, reducing the need for online improvement. We use *sparse reward* for experiments on Adroit.

**ManiSkill**. We evaluate on two ManiSkill (Gu et al., 2023; Mu et al., 2021; Tao et al., 2024) simulation tasks, *PushChair* and *TurnFaucet*. which require learning contact-rich manipulation with articulated objects. For training, we use demonstrations generated by model predictive control (for *TurnFaucet*) and by reinforcement learning policies (for *PushChair*). Because both data generation methods rely on dense reward functions, the resulting base Diffusion Policies exhibit minimal multimodality. We use *sparse reward* for experiments on ManiSkill.

### 5.1.2 BASE POLICY

We adopt **Diffusion Policy** (Chi et al., 2023) as our base multimodal policy. As a state-of-the-art imitation learning method, it generates robot action sequences via a conditional denoising diffusion process. Leveraging the power of diffusion-based generative models, Diffusion Policy is capable of effectively modeling multimodal behavior distributions. For fast inference and stable sample steering, we employ DDIM (Song et al., 2020) in diffusion sampling.

### 5.1.3 BASELINES

We compare our method against prior state-of-the-art fine-tuning and fine-tuning–free approaches.

**DSRL** (Wagenmaker et al., 2025) is an online RL method that optimizes the diffusion noise fed into a frozen Diffusion Policy, steering its sampler without updating network weights. However, because it constrains actions to the support of the base policy, its performance remains bounded by the quality and coverage of the demonstrations and the pre-trained model.

**Policy Decorator** (Yuan et al., 2024) is an online residual RL method that learns a residual policy, augmented with controlled exploration strategies such as bounded residual actions and a progressive exploration schedule, which provides a model-agnostic improvement over black-box base policies.

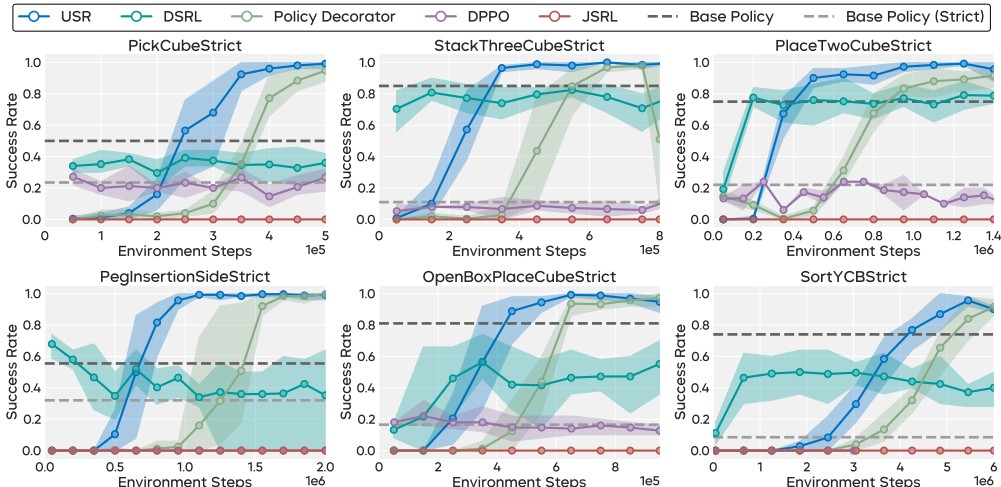

Figure 4: Learning curves on our proposed benchmark, **MultiModalBench** averaged over 5 seeds. The *Base Policy* line reports the success rate of the base Diffusion Policy when all behavior modes are counted as successful, whereas *Base Policy (Strict)* considers success only under a single designated behavior mode. Across all tasks, our method consistently outperforms the baseline methods.

**DPPO** (Ren et al., 2024) is an online RL method that finetunes a pre-trained Diffusion Policy using PPO (Schulman et al., 2017). By interacting with the environment, it incrementally adjusts the policy distribution to improve task performance.

**JSRL** (Uchendu et al., 2023) is an online curriculum learning method that leverages a base policy as a guiding policy. By using the base policy to steer the online policy toward the goal, JSRL reduces the difficulty of exploration and facilitates more efficient learning in complex tasks

## 5.2 EXPERIMENTAL RESULTS

**Our Method**. We evaluate USR on three benchmarks, including two standard manipulation benchmarks, Adroit (Rajeswaran et al., 2017) and ManiSkill (Gu et al., 2023), as well as on our proposed MultiModalBench. Tasks with the *Strict* suffix in MultiModalBench include multiple behavior modes in the demonstration, but only one is considered successful. This setting poses a significant challenge of steering the pretrained policy toward promising modes while refining actions to explore out-of-distribution area. As shown in Figure 4, USR significantly outperforms baselines, achieving both sample-efficient and near-perfect final performance. To test USR under more general and diverse conditions, we further evaluate on three tasks from the Adroit benchmark (Rajeswaran et al., 2017), using base policy model trained on human demonstrations. These demonstrations naturally induce implicit multimodal action distributions due to variability in human data collection. As shown in the top row of Figure 5, USR substantially outperforms baselines, highlighting its strength in utilizing human demonstrations. Finally, we evaluate on two tasks from the ManiSkill benchmark (Gu et al., 2023), where the base policy model are trained from demonstrations generated by Model Predictive Control and reinforcement learning policy learned under dense reward. These demonstrations are largely single-modal. As shown in the bottom row of Figure 5, USR consistently outperforms baselines, demonstrating its advantage even in settings with limited multimodality.

**Baselines**. We compare our method against a comprehensive set of baselines. As shown in Figure 4 and Figure 5, DSRL performs well on Adroit tasks with human demonstrations but struggles on MultiModalBench and ManiSkill tasks, which require either extra exploration besides mode steering or involve mostly single-modal demonstrations. These results suggest that while DSRL can quickly steer base actions toward promising modes, it lacks the ability to handle predominantly single-modal demonstrations or to achieve near-perfect performance beyond the base policy's support. More specifically, results on MultiModalBench show that DSRL can improve the base policy on *Strict* task to matches its performance on non-*Strict* tasks, indicating that DSRL is able to reach the correct behavior mode but cannot further boost performance beyond the base policy's support.

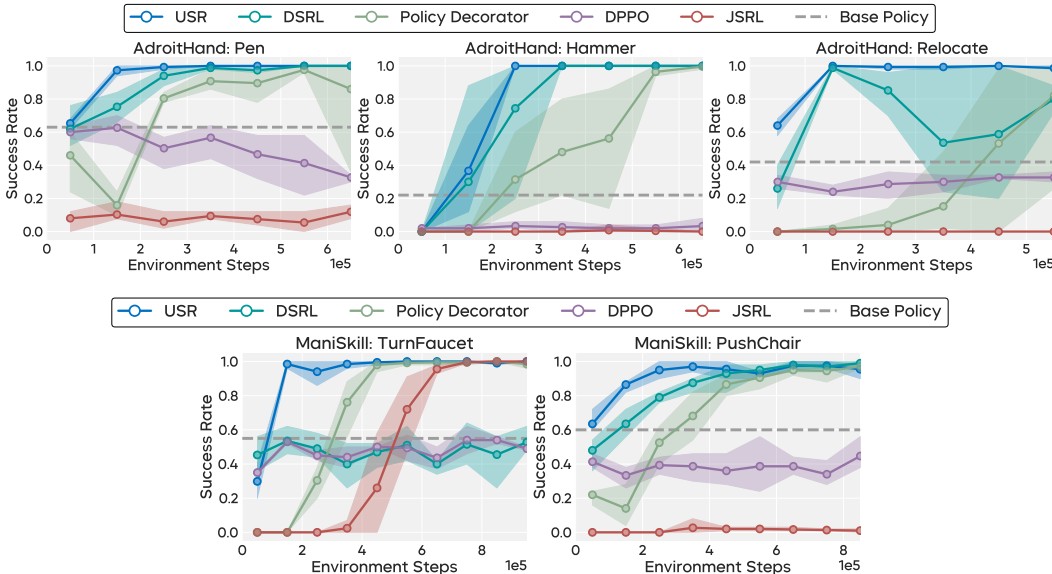

Figure 5: Learning curves on three **AdroitHand** tasks (top) and two **ManiSkill** tasks (bottom), averaged over 5 seeds. The *Base Policy* line reports the success rate of the base Diffusion Policy on that task.

We also find that DSRL suffers from limited training stability and is prone to collapse during training. In contrast, Policy Decorator provides stable and generally more reliable performance across all evaluated tasks. However, as it always treats the base policy as a black-box model, learning residual actions becomes considerably more difficult, and sample efficiency is reduced by the inability to leverage the base policy's output distribution. JSRL largely fails on MultiModalBench tasks but achieves some success on Adroit and ManiSkill due to exploration challenges. When the base policy falls into unwanted behavior modes, the student policy cannot make meaningful improvements without backtracking a long distance to the key decision state in order to select the intended behavior mode. Finally, we find that DPPO as an on-policy algorithm is considerably less sample-efficient than USR and incurs additional computational overhead.

**Visual Experiments**. We additionally evaluate USR with high-dimensional image observations. As shown in Appendix E.1, USR achieves superior performance over the baselines under visual inputs.

### 5.3 HYPERPARAMETER STUDY

We conduct hyperparameter studies on *OpenBoxPlaceCubeStrict* and *PlaceTwoCubeStrict* to provide further insights into the training dynamics of USR.

**Noise Magnitude** $b_w$. The hyperparameter $b_w$ controls the scale of the noise produced by the actor. As shown in Figure 6, we ablate $b_w$ over values ranging from 0.5 to 2.0 and observe similar performance across two tasks. These results suggest that $b_w$ is relatively insensitive to the choice of value. Following both our findings and the recommendation of the original paper (Wagenmaker et al., 2025), we set $b_w = 1.5$ for most experiments.

**Combined Critic Gradient Steps** $N_D$. The hyperparameter $N_D$ controls the number of combined critic updates performed to distill from environment critic in each training iteration. As shown in Figure 6, we ablate $N_D$ over values ranging from 1 to 8 and observe similar performance across two tasks. These results suggest that $N_D$ is relatively insensitive to the choice of value. Therefore, for training efficiency, we set $N_D = 1$ in most experiments.

**Residual Action Scale** $\alpha$. The hyperparameter $\alpha$ controls the maximum adjustment the residual policy can apply. As shown in Figure 6, a value that is too small leads to insufficient residual scaling, preventing the success rate from reaching 100%, whereas a value that is too large, such as $\alpha = 1.0$, significantly increases the difficulty of exploration, resulting in poor sample efficiency and even complete failure on the OpenBoxPlaceCubeStrict task. Across tasks, $\alpha$ demonstrates a generous

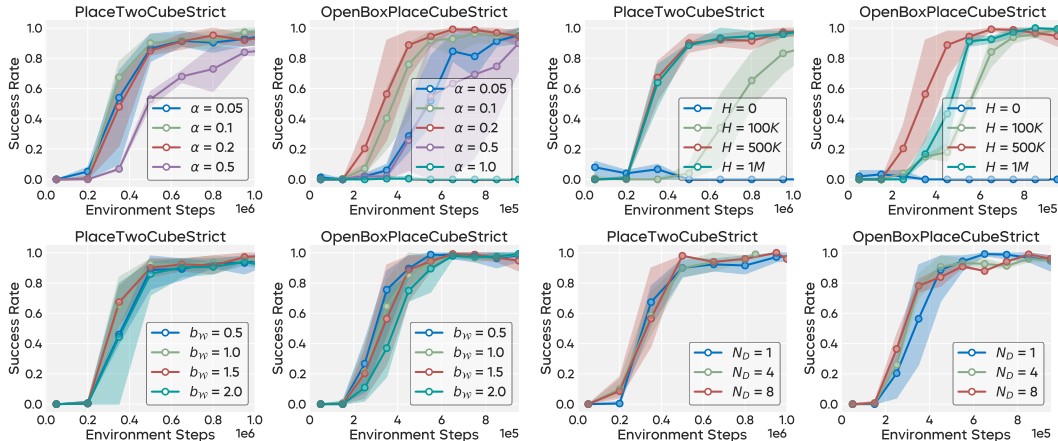

Figure 6: Ablations on four key hyperparameters on two tasks: residual scale $\alpha$, progressive exploration horizon $H$, noise magnitude $b_{\mathcal{W}}$, and combined critic update steps $N_D$.

workable range (0.1 to 0.5 for OpenBoxPlaceCubeStrict and 0.05 to 0.2 for PlaceTwoCubeStrict), making it comparatively easy to tune.

**Progressive Exploration Schedule $H$.** The hyperparameter $H$ controls the rate at which the policy switches from the base policy to the residual policy. As shown in Figure 6, a value that is too small, such as $H = 0$, increases the learning difficulty for the residual policy, resulting in reduced sample efficiency. In contrast, a larger $H$ is generally a safe choice.

## 5.4 Understanding USR

To better understand how our method USR achieves superior performance, we conduct additional qualitative studies to gain insights into the behavior of its two components. Specifically, we select an initial state from *PickCubeStrict*, sample the base policy 1000 times, apply PCA (Abdi & Williams, 2010) to project the actions, and plot the first principal component. We then apply the fully-trained USR to the base policy and visualize the first principal component of actions sampled from: (i) the base policy with noise provided by the unified actor, and (ii) the final actions after applying USR.

As shown in Figure 7, actions sampled directly from the base policy exhibit messy multimodal distributions, reflecting the multiple behavior modes inherent to the base policy. In contrast, actions sampled with noise provided by the unified actor form a clear single-modal distribution, effectively amplifying one pre-existing behavior mode. The final actions after applying USR preserve this single-modal structure while shifting the distribution

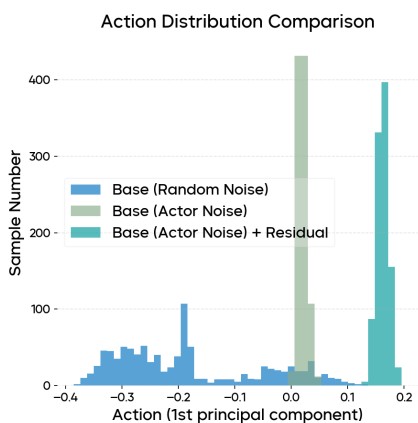

Figure 7: Action distribution comparison before and after USR.

along the $x$-axis. These observations suggest that the two components work together to improve the base policy more effectively: the noise action steers sampled trajectories toward the most promising mode, while the residual action enables further refinement beyond the support of the base policy.

## 5.5 Real Robot Experiments

To demonstrate the efficacy of USR in improving real robot policies, we conducted experiments on the Agibot G1 dual-arm platform. Unlike previous experiments that utilized standard diffusion policies, the base policy here is a multi-task Vision-Language-Action (VLA) model with a flow matching action expert. The model is trained on the AgiBot-World (Bu et al., 2025) dataset.

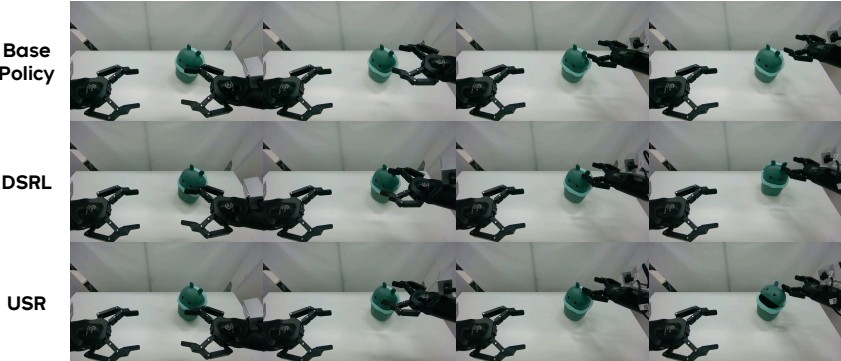

Figure 8: **Rollout comparison at a challenging position.** Base policy moves towards the bin but misses the interaction point. DSRL steers closer but acts too low, colliding with and displacing the bin. USR successfully refines the action to touch the lid's ears and completes the task.

**Task Description.** We focused on a fine-grained manipulation task: Lid Opening. The robot must use its right gripper to open a cartoon-styled trash bin placed on a tabletop. This task is challenging due to the required precision; the gripper must accurately catch and manipulate two small protruding "ears" on the lid to flip it open. A slight vertical misalignment results in the gripper colliding with the bin body, pushing the object away and causing task failure.

**Experimental Setup.** We benchmark the performance of the pre-trained VLA model as well as the improved policies by DSRL and USR. Throughout the experiments, the base VLA model was conditioned on a fixed language instruction: "open the lid of square trash bin with the right arm." Both DSRL and USR were trained online for 100 episodes. We employed a human-in-the-loop training protocol where a human supervisor provided a sparse binary reward (0/1) at the end of each episode and reset the object position when necessary.

**Results and Analysis.** The evaluation was conducted across 10 distinct object positions with 2 trials per position (20 evaluation episodes in total). As reported in Table 1, the pre-trained base VLA model achieved a success rate of 40% (8/20). DSRL improved performance to 75% (15/20) through latent steering, while USR achieved the highest success rate of 90% (18/20). To investigate the underlying causes of this performance gap, we visualized rollouts at a particular position where both the base

Table 1: Success rates on the Lid Opening task.

| Method | Success / Total |
| --- | --- |
| Base Policy | 8 / 20 |
| DSRL | 15 / 20 |
| **USR** | **18 / 20** |

model and DSRL failed (see Figure 8). The base VLA model exhibited the correct general intent by moving the right gripper toward the bin but failed to make effective contact with the lid due to a lack of precision. DSRL successfully steered the gripper closer to the target; however, it executed the grasp slightly too low, causing the gripper to push the bin body rather than opening the lid. In contrast, USR successfully leveraged its residual component to apply a fine-grained vertical correction, allowing the gripper to precisely align with the lid's "ears" and successfully flick it open.

The real robot results validate that USR is compatible with state-of-the-art VLA architectures and confirm that the residual refinement module is critical for achieving fine-grained manipulation tasks that are difficult to solve via latent steering alone.

## 6 CONCLUSION

We introduce Unified latent Steering and residual Refinement (USR), a novel framework for the online improvement of diffusion policy models. USR utilizes a lightweight actor to jointly steer the diffusion process with latent noise and apply residual corrections to the sampled action. This unified design combines stable mode selection with flexible adaptation, overcoming the limitations of prior methods. Experiments on our new MultiModalBench, along with Adroit and ManiSkill benchmarks, show that USR achieves state-of-the-art performance and sample efficiency by effectively selecting promising behavioral modes and refining actions beyond the base policy's support.

## CODE OF ETHICS

We affirm that our research adheres to the ethical guidelines set forth in the ICLR Code of Ethics. We have ensured the integrity of our data, transparency in results, and compliance with all applicable laws and regulations. Our research does not involve human subjects. We also disclose any potential conflicts of interest and strive for fairness and non-discrimination in our work.

## REPRODUCIBILITY STATEMENT

We are committed to making our results reproducible. The source code and datasets will be made publicly available upon publication. All experimental details, including hyperparameters, model configurations, and evaluation metrics, are documented clearly in the paper and supplementary materials.

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

# A   DECLARATION OF LLM USAGE

Large Language Models (LLMs) were used in the preparation of this submission. Specifically, they assisted in editing and polishing the writing for grammar and clarity. All technical ideas, experimental designs, and results were developed by the authors.

# B   ALGORITHM SUMMARY

The complete USR algorithm is summarized in Algorithm 1. The process involves collecting experience using the unified steering and refinement mechanism, and then updating the two critics and the actor using data from the replay buffer.

---

**Algorithm 1** Unified Latent Steering and Residual Refinement (USR)

---

1: **Initialize:** Unified actor $\pi_\theta$, critics $Q_\phi^{\mathcal{A}}$, $Q_\psi^{\mathcal{C}}$, target networks, replay buffer $\mathcal{D}$, residual scale $\alpha$, progressive exploration horizon $H$.
2: **Load** pretrained, frozen diffusion policy $\pi_{\mathrm{dp}}$.
3: **for** each timestep $t = 1, \ldots, T$ **do**
4:     Observe state $o_t$.
5:     Sample combined action $a_t^{\mathrm{comb}} = [w_t, a_t^{\mathrm{res}}] \sim \pi_\theta(\cdot|o_t)$.
6:     Steer base policy to get intermediate action: $\tilde{a}_t = \pi_{\mathrm{dp}}(o_t, w_t)$.
7:     Calculate exploration probability $\epsilon = \min(t/H, 1.0)$.
8:     **if** Uniform(0,1) $< \epsilon$ **then**
9:         Refine action: $a_t = \tilde{a}_t + \alpha \cdot a_t^{\mathrm{res}}$.
10:     **else**
11:         Use steered base action only: $a_t = \tilde{a}_t$.
12:     Execute $a_t$, observe reward $r_t$ and next observation $o_{t+1}$.
13:     Store transition $(o_t, a_t, r_t, o_{t+1})$ in replay buffer $\mathcal{D}$.
14:     **for** each gradient step **do**
15:         Sample minibatch of transitions from $\mathcal{D}$.
16:         **Update Environment Critic $Q_\phi^{\mathcal{A}}$:**
17:             Compute TD targets $y$ and update $\phi$ to minimize $\mathcal{L}_{\mathrm{TD}}(\phi)$.
18:         **Update Combined Critic $Q_\psi^{\mathcal{C}}$:**
19:             Sample observations $o$ and random combined actions $a^{\mathrm{comb}} = [w, a^{\mathrm{res}}]$.
20:             Compute target values $Q_\phi^{\mathcal{A}}(o, \pi_{\mathrm{dp}}(o, w) + \alpha \cdot a^{\mathrm{res}})$.
21:             Update $\psi$ to minimize the distillation loss $\mathcal{L}_{\mathrm{distill}}(\psi)$.
22:         **Update Actor $\pi_\theta$:**
23:             Update $\theta$ by ascending the policy gradient from the SAC objective using $Q_\psi^{\mathcal{C}}$.
24:         Update target networks.

---

# C   FURTHER DETAILS ON THE EXPERIMENTAL SETUP

## C.1   TASK DESCRIPTIONS

We consider a total of 11 continuous robotic control tasks from 3 benchmarks: our proposed Multi-ModalBench, Adroit (Rajeswaran et al., 2017), and ManiSkill (Gu et al., 2023; Mu et al., 2021; Tao et al., 2024). This section provides detailed task descriptions on overall information, task difficulty, object sets, state space, and action space. Refer to Table 2 for detailed information.

### C.1.1   MULTIMODALBENCH TASKS

For MultiModalBench, tasks without the *Strict* suffix count all behavior modes as successful, whereas tasks with the *Strict* suffix only regard a single designated behavior mode as success. Our evaluation focuses on the *Strict* variants, where the goal is to maximize performance under this stricter success criterion. Refer to Figure 9 for detailed illustration of different modes.

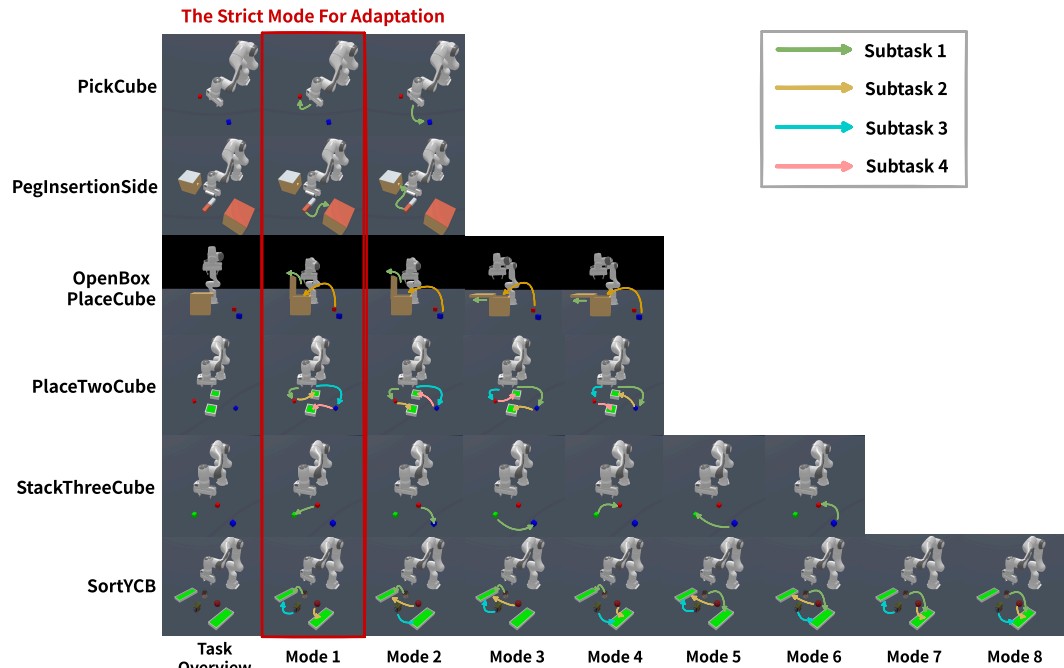

Figure 9: Visualization of all behavior modes for the six tasks in MultiModalBench. Each row corresponds to a task, with columns depicting distinct demonstration modes present in the dataset. For tasks marked with the *Strict* suffix, only one mode (outlined in red) is considered the adaptation target, while the others are only included in demonstrations. These visualizations illustrate the multimodal nature of the base policies training on the demonstration, highlighting the challenge of steering the pretrained policy toward the correct mode while refining actions to achieve precise task success.

**PickCube/PickCubeStrict**

- Overall Description: One red cube and one blue cube are placed on the table. The task is to pick up one cube, while the *Strict* variant requires specifically picking up the red cube.
- Task Difficulty: The two cubes are placed at randomized positions within a designated region of the table.
- Object Variations: No Object Variations.
- Action Space: Delta position of the end-effector and joint positions of the gripper.
- State Observation Space: Proprioceptive robot state information, such as joint angles and velocities of the robot arm, and task-specific goal information, which is represented by the poses of the two cubes.
- Visual Observation Space: One 64x64 RGBD image from a base camera and one 64x64 RGBD image from a hand camera.

**PegInsertionSide/PegInsertionSideStrict**

- Overall Description: One peg and two holes are placed on the table. The task is to insert the peg into either hole, while the *Strict* variant requires inserting it into a designated hole.
- Task Difficulty: The peg and two holes are placed at randomized positions within a designated region of the table.
- Object Variations: No Object Variations.
- Action Space: Delta position of the end-effector and joint positions of the gripper.
- State Observation Space: Proprioceptive robot state information, such as joint angles and velocities of the robot arm, and task-specific goal information.

- Visual Observation Space: Visual Observation Space: One 64x64 RGBD image from a base camera and one 64x64 RGBD image from a hand camera.

**OpenBoxPlaceCube/OpenBoxPlaceCubeStrict**

- Overall Description: A box with a cover and two cubes are placed on the table. The task is to choose a cube and place it inside the box by either sliding or lifting the cover, while the *Strict* variant requires lifting the cover and pick the red cube.
- Task Difficulty: The box and the cubes are placed at randomized positions within a designated region of the table.
- Object Variations: No Object Variations
- Action Space: Delta position of the end-effector and joint positions of the gripper.
- State Observation Space: Proprioceptive robot state information, such as joint angles and velocities of the robot arm, and task-specific goal information.
- Visual Observation Space: Visual Observation Space: One 64x64 RGBD image from a base camera and one 64x64 RGBD image from a hand camera.

**PlaceTwoCube/PlaceTwoCubeStrict**

- Overall Description: Two cubes and two boxes are placed on the table. The task is to place each cube into a separate box, while the *Strict* variant requires first placing the red cube into box 1 and then placing the blue cube into box 2.
- Task Difficulty: The two cubes and two boxes are placed at randomized positions within a designated region of the table.
- Object Variations: No Object Variations.
- Action Space: Delta position of the end-effector and joint positions of the gripper.
- State Observation Space: Proprioceptive robot state information, such as joint angles and velocities of the robot arm, and task-specific goal information.
- Visual Observation Space: Visual Observation Space: One 64x64 RGBD image from a base camera and one 64x64 RGBD image from a hand camera.

**StackThreeCube/StackThreeCubeStrict**

- Overall Description: Three cubes are placed on the table. The task is to select one cube to be placed on top of another, while the *Strict* variant requires stacking the red cube on top of the green cube.
- Task Difficulty: The three cubes are placed at randomized positions within a designated region of the table.
- Object Variations: No Object Variations
- Action Space: Delta position of the end-effector and joint positions of the gripper.
- State Observation Space: Proprioceptive robot state information, such as joint angles and velocities of the robot arm, and task-specific goal information.
- Visual Observation Space: Visual Observation Space: One 64x64 RGBD image from a base camera and one 64x64 RGBD image from a hand camera.

**SortYCB/SortYCBStrict**

- Overall Description: Three YCB objects (Calli et al., 2015) and two boxes are placed on the table. The task is to place the objects into the boxes, while the *Strict* variant requires placing them into designated boxes.The placement order of objects is fixed, while the choice of boxes is randomized.
- Task Difficulty: The objects and boxes are placed at randomized positions within a designated region of the table, and the objects exhibit shape variations.

- Object Variations: The objects exhibit shape variations.

- Action Space: Delta position of the end-effector and joint positions of the gripper.

- State Observation Space: Proprioceptive robot state information, such as joint angles and velocities of the robot arm, and task-specific goal information.

- Visual Observation Space: Visual Observation Space: One 64x64 RGBD image from a base camera and one 64x64 RGBD image from a hand camera.

### C.1.2 ADROITHAND TASKS

We experiment with three simulation tasks from the AdroitHand benchmark (Rajeswaran et al., 2017): **Pen**, **Hammer**, and **Relocate**. We exclude the **Door** task, as the base Diffusion Policy already achieves near-perfect performance with the demonstrations.

**Pen**

- Overall Description: The environment is based on the Adroit manipulation platform, a 28 degree of freedom system which consists of a 24 degrees of freedom ShadowHand and a 4 degree of freedom arm. The task to be completed consists on repositioning the blue pen to match the orientation of the green target.

- Task Difficulty: The target is randomized to cover all configurations.

- Object Variations: No Object Variations.

- Action Space: Absolute angular positions of the Adroit hand joints.

- State Observation Space: The angular position of the finger joints, the pose of the palm of the hand, as well as the pose of the real pen and target goal.

**Hammer**

- Overall Description: The environment is based on the Adroit manipulation platform, a 28 degree of freedom system which consists of a 24 degrees of freedom ShadowHand and a 4 degree of freedom arm. The task to be completed consists on picking up a hammer with and drive a nail into a board.

- Task Difficulty: The nail position is randomized and has dry friction capable of absorbing up to 15N force.

- Object Variations: No Object Variations.

- Action Space: Absolute angular positions of the Adroit hand joints.

- State Observation Space: The angular position of the finger joints, the pose of the palm of the hand, the pose of the hammer and nail, and external forces on the nail.

**Relocate**

- Overall Description: The environment is based on the Adroit manipulation platform, a 30 degree of freedom system which consists of a 24 degrees of freedom ShadowHand and a 6 degree of freedom arm. The task to be completed consists on moving the blue ball to the green target.

- Task Difficulty: The positions of the ball and target are randomized over the entire workspace.

- Object Variations: No Object Variations.

- Action Space: Absolute angular positions of the Adroit hand joints.

- State Observation Space: The angular position of the finger joints, the pose of the palm of the hand, as well as kinematic information about the ball and target.

### C.1.3 MANISKILL TASKS

**PushChair**

- Overall Description: The environment is based on a dual-arm manipulation setup. The task requires the robot to make contact with a chair and push it to a designated target location on the ground.

- Task Difficulty: The initial pose of the chair and the goal location are randomized, requiring robust coordination of both arms to achieve stable pushing.

- Object Variations: There are 10 different chairs.

- Action Space: End-effector delta position and rotation commands for both arms, together with gripper control.

- State Observation Space: The joint angles and velocities of both arms, the poses of the two end-effectors, and the pose of the chair and its goal position.

**TurnFaucet**

- Overall Description: The environment is based on a 7 degree of freedom single-arm robot. The task requires the robot to grasp and rotate a faucet handle to a target angle.

- Task Difficulty: The initial pose of the faucet is randomized, and successful completion requires precise manipulation to overcome torque and resistance at the joint.

- Object Variations: There are 10 different faucets.

- Action Space: End-effector delta position and rotation commands, together with gripper open/close control.

- State Observation Space: The joint angles and velocities of the robot arm, the end-effector pose, and the pose of the faucet including its current and goal angles.

Table 2: We list important task details below.

| Task | State Observation Dim | Action Dim | Max Episode Steps |
|---|---|---|---|
| PickRedCubeStrict | 52 | 7 | 150 |
| PegInsertionSideStrict | 57 | 7 | 200 |
| OpenBoxPlaceCubeStrict | 63 | 7 | 400 |
| PlaceTwoCubeStrict | 53 | 7 | 600 |
| StackThreeCubeStrict | 62 | 7 | 200 |
| SortYCBStrict | 68 | 7 | 650 |
| | | | |
| AdroitHandPen | 46 | 24 | 200 |
| AdroitHandHammer | 46 | 26 | 400 |
| AdroitHandRelocate | 39 | 30 | 400 |
| | | | |
| TurnFaucet | 43 | 7 | 200 |
| PushChair | 131 | 20 | 200 |

## C.2 DEMONSTRATIONS

This section provides the details of demonstrations used in our experiments. Refer to Table 3 for detailed information.

Table 3: We list the number of demonstrations and corresponding generation methods below

| Task | Num of Demo Trajs | Generation Method |
|---|---|---|
| PickRedCubeStrict | 200 | Task and Motion Planning (TAMP) |
| PegInsertionSideStrict | 1000 | Task and Motion Planning (TAMP) |
| OpenBoxPlaceCubeStrict | 200 | Task and Motion Planning (TAMP) |
| PlaceTwoCubeStrict | 200 | Task and Motion Planning (TAMP) |
| StackThreeCubeStrict | 100 | Task and Motion Planning (TAMP) |
| SortYCBStrict | 1000 | Task and Motion Planning (TAMP) |
| AdroitHandPen | 25 | Human Demonstrations |
| AdroitHandHammer | 25 | Human Demonstrations |
| AdroitHandRelocate | 25 | Human Demonstrations |
| TurnFaucet | 1000 | Model Predictive Control (MPC) |
| PushChair | 1000 | Reinforcement Learning (RL) |

# D    IMPLEMENTATION DETAILS

## D.1    BASE POLICY

We experiment with state-of-the-art diffusion-based imitation learning methods, Diffusion Policy (Chi et al., 2023) for all of our experiments.

### D.1.1    DIFFUSION POLICY

We follow the setup of U-Net (Ronneberger et al., 2015) version of Diffusion Policy in the original paper (Chi et al., 2023).

Table 4: We list the important architecture hyperparameters of Diffusion Policy used in our experiments.

| Hyperparamter | Value (MultiModalBench) | Value (Adroit) | Value (ManiSkill) |
|---|---|---|---|
| Observation Horizon | 2 | 2 | 2 |
| Action Horizon | 4 | 4 | 4 |
| Prediction Horizon | 16 | 16 | 16 |
| Embedding Dimensions | 64 | 64 | 64 |
| Downsampling Dimensions | 256, 512, 1024 | 256, 512, 1024 | 256, 512, 1024 |
| Trainable Parameters | About 4 Million | About 4 Million | About 4 Million |

Table 5: We list the important training hyperparameters of Diffusion Policy used in our experiments.

| Hyperparameter | Value (MultiModalBench) | Value (Adroit) | Value (ManiSkill) |
|---|---|---|---|
| Gradient Steps | 200000 | 200000 | 200000 |
| Batch Size | 1024 | 1024 | 1024 |
| Learning Rate | 1e-4 | 1e-4 | 1e-4 |
| Optimizer | AdamW Optimizer | AdamW Optimizer | AdamW Optimizer |

### D.1.2    CHECKPOINT SELECTION

We evaluate the base policy for 50 episodes every 5k training steps during training. We select the checkpoint at a fixed step after the convergence of the base policy.

## D.2 USR (OUR METHOD)

Our method USR involves four algorithm-specific hyperparameters and a set of shared hyperparameters, as introduced in Section 4.1. Detailed descriptions of the algorithm-specific hyperparameters are provided in Section D.2.1, and the shared hyperparameters are summarized in Section D.2.2.

### D.2.1 USR SPECIFIC HYPERPARAMETERS

As introduced in Section. 4.1, our method includes four algorithm-specific hyperparameters. Detailed information is provided in Table 6. For fair comparison, the DSRL baseline uses the same $b_w$ and $N_D$, while the Policy Decorator baseline uses the same $\alpha$ and $H$ as USR.

Table 6: We list USR specific hyperparameters below.

| Task | $b_w$ | $N_D$ | $\alpha$ | $H$ |
|---|---|---|---|---|
| PickCubeStrict | 1.5 | 1 | 0.2 | 300K |
| PegInsertionSideStrict | 1.5 | 1 | 0.1 | 300K |
| OpenBoxPlaceCubeStrict | 1.5 | 1 | 0.2 | 500K |
| PlaceTwoCubeStrict | 1.5 | 1 | 0.1 | 500K |
| StackThreeCubeStrict | 1.5 | 1 | 0.1 | 500K |
| SortYCBStrict | 0.5 | 4 | 0.3 | 800K |
| AdroitHandPen | 1.5 | 1 | 0.2 | 0 |
| AdroitHandHammer | 1.5 | 1 | 0.05 | 0 |
| AdroitHandRelocate | 1.5 | 1 | 0.1 | 300K |
| TurnFaucet | 1.5 | 1 | 0.1 | 100K |
| PushChair | 1.5 | 1 | 0.2 | 300K |

### D.2.2 IMPORTANT SHARED HYPERPARAMETERS

Table 7: We list important shared hyperparameters below.

| Hyperparameter | Value |
|---|---|
| Gamma | 0.97 |
| Batch Size | 1024 |
| Learning Rate | 1e-4 |
| Policy Update Frequencey | 1 |
| Training Frequency | 64 |
| UTD Ratio | 0.25 |
| Target Network Update Frequency | 1 |
| Tau | 0.01 |
| Learning Starts | 8000 |

There are several shared hyperparameters of the SAC algorithm (Haarnoja et al., 2018) that are used across multiple baselines. Although the DSRL paper recommends using a high UTD, it also acknowledges that UTD is highly environment-specific. In our tasks, we find that high UTD leads to either significant training instability or only minimal gains. Therefore, for both fair comparison and training efficiency, we adopt the same UTD values for the DSRL baseline as for the other methods. Refer to Table 7 for detailed information.

### D.2.3 ACTOR AND CRITIC ARCHITECTURES

The unified actor network consists of a three-layer MLP, where the first layer takes as input the observation vector and each hidden layer has a dimension of 2048. The actor further includes two

additional MLP heads: a mean-action head and a standard-deviation head. Both heads take a 2048-dimensional input and output vectors matching the action dimension, following the standard SAC actor design. A ReLU activation is applied after every MLP layer.

The combined critic and environment critic share the same architecture except for their input dimensions. Each critic is a three-layer MLP with hidden dimension 2048. The combined critic takes as input the concatenation of the observation and both the noise and residual actions, resulting in an input dimension of obs_dim + 2 × act_dim. The environment critic takes the observation and the executed action as input, giving an input dimension of obs_dim + act_dim. A ReLU activation is applied after every MLP layer.

# E  ADDITIONAL EXPERIMENTAL RESULTS

This section includes some additional experiments. Section E.1 includes visual observation experiments.

## E.1  VISUAL OBSERVATION EXPERIMENTS

We evaluate USR with high-dimensional visual observations. As shown in Figure 10, USR achieves superior performance over the baselines under visual inputs.

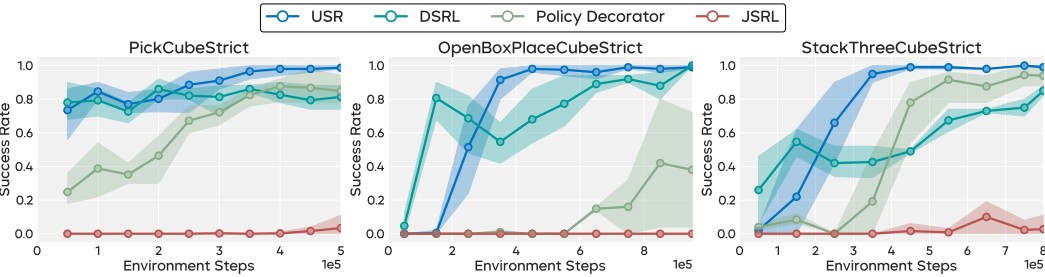

Figure 10: Learning curves on three **MultiModalBench** environments using image observations averaged over 5 seeds. We omit the remaining tasks because the base Diffusion Policy fails to achieve a reasonable success rate under visual inputs.

## E.2  ROBOMIMIC EXPERIMENTS

To more comprehensively evaluate the performance and generality of our proposed method, **USR**, across diverse robotic tasks and human-generated datasets—and to enable direct comparison with prior baselines on their evaluated settings—we additionally conduct experiments on two representative **RoboMimic** tasks: **can** and **square**. We choose these tasks because **lift** is relatively trivial for the base policy, whereas **transport** requires excessive computational resources.

We further note that the original DSRL paper (Wagenmaker et al., 2025) evaluates its method using base diffusion policies with relatively high success rates (approximately 70%). Such a setting creates a sizable optimal region within the base policy's action distribution, implicitly favoring noise-steering approaches like DSRL. We consider this setup unrepresentative of more realistic scenarios where pre-trained policies are imperfect. To stress-test USR's ability to improve policies online, we intentionally train a weak base policy. This yields a more challenging regime in which the pre-trained distribution does not reliably cover successful executions, forcing the adaptation method to explore beyond the initial support.

As shown in the newly added Figure 11, the results clearly reveal the limitations of DSRL and underscore the necessity of USR. On the **can** task, DSRL—which performs pure noise steering—fails to achieve any meaningful improvement. On **square**, its performance displays extremely high variance, with only a small fraction of seeds improving upon the base policy. We hypothesize that this failure stems from the intrinsic limitations of noise steering: although the pre-trained distribution

may include the correct behavioral mode, its actions lack the precision required to reliably obtain the sparse rewards.

In contrast, **USR consistently improves the success rate to nearly 100% on both tasks**, demonstrating strong stability and robustness. This large performance gap highlights the importance of incorporating a residual component that enables the policy to refine actions and effectively explore beyond the base policy's support. By allowing controlled out-of-distribution corrections, USR bridges the gap between an imperfect pre-trained policy and the precision necessary for reliable task completion.

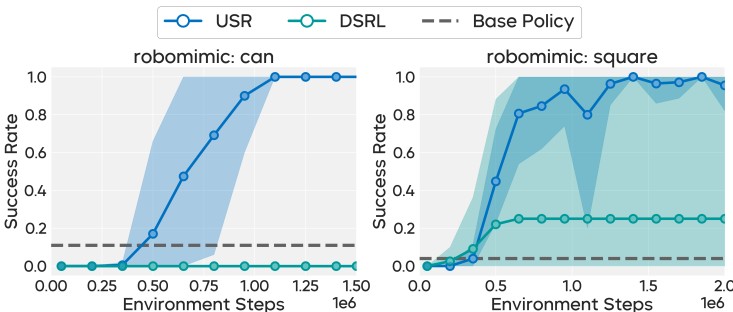

Figure 11: Learning curves on two **RoboMimic** environments using state observations averaged over 5 seeds.

### E.3 ABLATION STUDY ON DUAL-CRITIC VS. SINGLE-CRITIC ARCHITECTURES

It is worth noting that the original DSRL paper (Wagenmaker et al., 2025) introduces a simpler variant of the algorithm, termed **DSRL-SAC**, which employs a single combined critic rather than the dual-critic architecture used in the **DSRL-NA** version (where *NA* denotes noise aliasing). Their paper argues that the noise-aliasing formulation reduces unnecessary exploration in the latent-noise space and consequently improves sample efficiency. This motivates our choice to adopt the noise-aliasing version as the primary configuration for USR. To more comprehensively evaluate our method, we additionally conduct ablation studies comparing the **single-critic architecture** (SAC variants) and the **dual-critic architecture** (NA variants).

As shown in Figure 12, both **DSRL-SAC** and **USR-SAC** underperform their corresponding noise-aliasing variants (**DSRL-NA** and **USR-NA**) on the MultiModalBench *PickCubeStrict* and Adroit *Pen* tasks. These results align with the observations reported in the original DSRL paper and further justify our algorithmic design choice of using the noise-aliasing formulation.

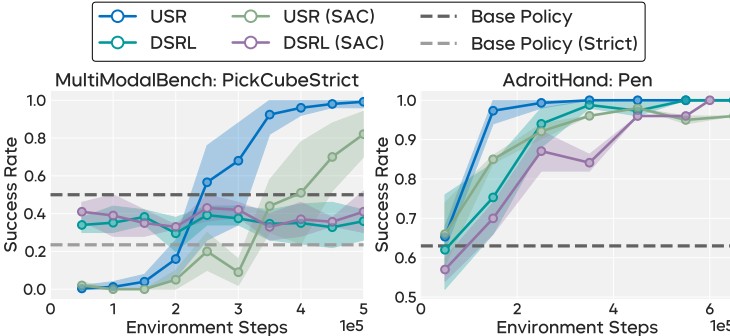

Figure 12: Learning curves for the ablation experiments comparing single-critic and dual-critic architectures on the **PickCubeStrict** and **Pen** environments using state observations, averaged over 5 seeds.

### E.4 USR WITH OFFLINE DEMONSTRATIONS

It is a common technique in offline-to-online RL to leverage offline demonstrations in the online replay buffer to improve sample efficiency, as discussed in Wagenmaker et al. (2025); Nakamoto et al. (2023). To more comprehensively evaluate our method, we conduct additional experiments that incorporate offline demonstrations into USR's replay buffer. As shown in Figure 13, integrating offline demonstrations does not provide a noticeable improvement in sample efficiency.

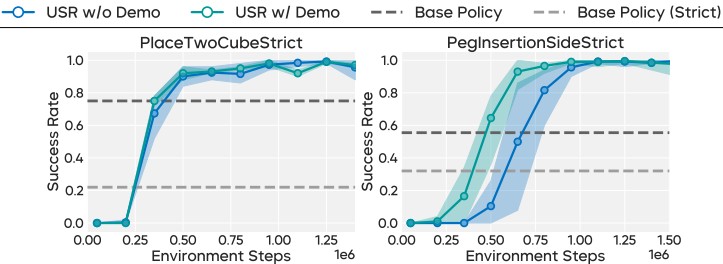

Figure 13: Learning curves for the experiments that incorporate offline demonstrations into the online replay buffer of USR on the **PlaceTwoCubeStrict** and **PegInsertionSideStrict** environments using state observations, averaged over 5 seeds.

### E.5 EXPERIMENTS ON IMBALANCED MODE DATASETS

The dataset used in our main **MultiModalBench** experiments, as shown in Figure 4, contains an equal number of demonstrations for each behavior mode. To more comprehensively evaluate our method under challenging data distributions, we additionally conduct experiments where the expected mode is underrepresented in the dataset.

As shown in Figure 14, our method maintains robust performance in different levels of the underrepresented (10% and 30%) settings, whereas DSRL experiences substantial degradation when the expected mode is underrepresented.

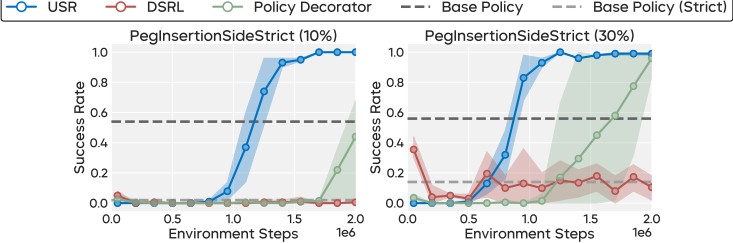

Figure 14: Learning curves for the experiments using imbalanced mode datasets on the **PegInsertionSideStrict** environment with state observations, averaged over 5 seeds.

### E.6 HYPERPARAMETER STUDY ON DISCOUNT FACTOR (GAMMA)

We follow several principles when selecting the discount factor. First, we use a consistent value within each benchmark to avoid excessive tuning. Second, whenever possible, we adopt discount factors used in prior work to ensure fair comparison with existing baselines. Third, all methods in our experiments use the same discount factor to maintain fairness across approaches.

We additionally perform a sweep over different discount factors on **MultiModalBench** and **Adroit**. The results in Figure 15 show that a value of 0.97 performs best on **MultiModalBench**, whereas all tested values yield similar performance on **Adroit**, supporting our final choice of discount factor.

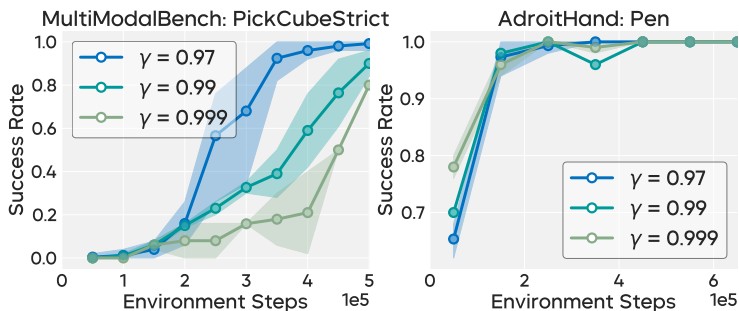

Figure 15: Learning curves for the experiments with different discount factor values on the **Pick-CubeStrict** and **Pen** environment with state observations, averaged over 5 seeds.

## E.7 HYPERPARAMETER STUDY ON UPDATE-TO-DATA RATIO (UTD)

The Update-to-Data Ratio (UTD) specifies how many gradient update steps are performed per environment timestep of collected data. Increasing the UTD can improve sample efficiency, but typically comes at the cost of substantially longer training time. Our choice of UTD for each benchmark is therefore made to balance sample efficiency and wall-clock runtime.

We also conducted a hyperparameter study on the UTD. As shown in Figure 16, setting UTD = 1.0 yields only marginal gains in sample efficiency while significantly slowing down training. Consequently, we select a UTD value that best trades off sample efficiency against training speed.

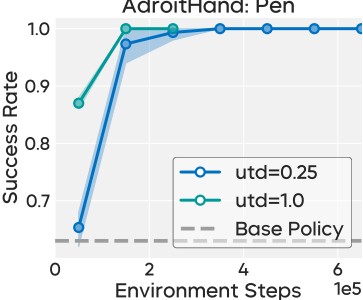

Figure 16: Learning curves for the experiments with different UTD values on the **Pen** environment with state observations, averaged over 5 seeds.

