# OpenReview forum: "Unified Latent Steering and Residual Refinement for Online Improvement of Diffusion Policy Models"
_ICLR.cc/2026/Conference — Submitted to ICLR 2026_

### Official Review · Reviewer_Ke1h · 2025-10-30

**Soundness:** 3
**Presentation:** 3
**Contribution:** 2
**Rating:** 4
**Confidence:** 5

**Summary:**

This paper presents USR, a lightweight framework for online improvement of pretrained diffusion robot manipulation policies. It combines latent noise steering to exploit multimodal base policy behaviors and residual corrections to enable adaptation beyond base policy support. Experiments on MultiModalBench, a new benchmark for evaluating multimodal manipulation, AdroitHand and ManiSkill show USR outperforms strong baselines in sample efficiency and task success.

**Strengths:**

- This paper is overall clearly written.
- The experiments cover a range of test settings, and the ablation studies help understand each component of the method.
- Overall the experiment results are good, which demonstrates the effectiveness of the proposed method.

**Weaknesses:**

**Weakness 1: Real-world validation is absent.**

There are no real-robot experiments to validate the effectiveness of the proposed method in policy learning in the real world. Incorporating such experiments would provide a more comprehensive understanding of USR's performance in real-world policy learning.


**Weakness 2: The analysis of USR’s performance in long-horizon tasks is insufficient.**

The maximum episode steps across all experiments are only 650 (for SortYCBStrict), and there is no validation of the method’s adaptability to longer-cycle manipulation tasks (e.g., multi-step assembly, continuous object transportation) where cumulative errors or mode drift might occur.

**Weakness 3: Limited discussion of constraints and theoretical failure cases:**

The paper notes the actor-critic mechanism but does not delve into what happens if the critic overfits to spurious correlations early in adaptation, or how USR mitigates bounce-back or drifting due to over-reliance on the residual path.

**Questions:**

**Question 1:**

Can the authors clarify how the dimensions and sampling distributions for $ w_t $ are chosen and normalized relative to the base diffusion model? Is there any adaptive mechanism (e.g., temperature, scale) for action/noise selection beyond a static bound?

**Question 2:**

How does USR perform in settings in which demonstration coverage is incomplete or severely imbalanced, e.g., where certain modes are missing or underrepresented? Are there experiments quantifying such regime limitations?

---

> ### Author Response · Authors · 2025-12-03
>
> **Q1: "Real-world validation is absent."**
>
> **A1:** We thank the reviewer for this valuable suggestion. We have added real-world experiments on the Agibot G1 platform (see Section 5.5). Crucially, to demonstrate scalability, we applied USR to fine-tune a large-scale pre-trained Vision-Language-Action model on a "Lid Opening" task.
>
> As shown in Table 1, while the Base VLA model (40% success) and DSRL (75% success) struggled with precise object interaction, USR achieved a 90% success rate. This validates that USR is not limited to simulation but is capable of efficiently improving behavioral foundation models in the physical world, confirming its practical applicability.
>
> **Q2: "The analysis of USR’s performance in long-horizon tasks is insufficient."**
>
> **A2:** We believe that the maximum timestep is not a decisive indicator of whether a task is long-horizon. Rather, it is an environment hyperparameter introduced to avoid unnecessary transitions and improve sample efficiency. For instance, one could trivially set the maximum timestep of a simple PickCube task to 1000, but this does not make the task long-horizon. Instead, long-horizon complexity arises from multi-stage manipulation sequences, which our benchmarks already capture. For example, OpenBoxPlaceCube requires opening the box, grasping the cube, and placing it inside, while SortYCB involves grasping and placing three different objects in sequence. These multi-step dependencies reflect genuine long-horizon structure. Therefore, we believe our evaluations sufficiently validate performance on long-horizon tasks.
>
> **Q3: "Limited discussion of constraints and theoretical failure cases"**
>
> **A3:** Critic overfitting to spurious correlations is typically an issue in offline RL rather than online RL, as online interaction provides sufficient fresh data to correct such errors if they arise. Beyond this inherent robustness, our design explicitly mitigates the risks of critic overfitting and residual-path drift. The dual-critic structure (Section 4.2) stabilizes value estimation by distilling the combined critic from the more reliable environment critic, preventing the actor from being influenced by spurious correlations early in training. Moreover, the frozen diffusion base policy anchors behavior within its multimodal support, while the residual actions are bounded and gradually introduced through a progressive exploration schedule. Together, these mechanisms prevent over-reliance on the residual path and avoid the bounce-back or drifting behaviors seen in prior residual-only approaches.
>
> **Q4: "Can the authors clarify how the dimensions and sampling distributions for  are chosen and normalized relative to the base diffusion model? Is there any adaptive mechanism (e.g., temperature, scale) for action/noise selection beyond a static bound?"**
>
> **A4:** The dimension of the noise vector matches the prediction horizon of the base diffusion policy. In standard diffusion policies, this noise is sampled from a standard Gaussian distribution. In both our method and DSRL, however, we replace this with the output of the noise actor (or the noise component of the unified actor) as the initial noise. Aside from applying a fixed magnitude bound to this noise, our method does not introduce any additional adaptive mechanisms.
>
> **Q5: "How does USR perform in settings in which demonstration coverage is incomplete or severely imbalanced, e.g., where certain modes are missing or underrepresented? Are there experiments quantifying such regime limitations?"**
>
> **A5:** The dataset used in our main MultiModalBench experiments, as shown in Figure 4, contains an
> equal number of demonstrations for each behavior mode. To more comprehensively evaluate our
> method under diverse data distributions, we additionally conduct experiments in two scenarios where
> the expected mode is underrepresented in the dataset.
> As shown in Figure 14, our method maintains robust performance in different levels of underrepresented
> (10% and 30%) settings, whereas DSRL experiences substantial degradation when
> the expected mode is underrepresented. Check Appendix E.5 for more detailed information.

---

### Official Review · Reviewer_2Zek · 2025-11-01

**Soundness:** 2
**Presentation:** 3
**Contribution:** 2
**Rating:** 4
**Confidence:** 3

**Summary:**

This paper proposes an unified framework for online fine-tuning diffusion policies with a combination of sample steering and residual refinement. The idea is simple and straightforward with reasonably good performance on three simulation benchmarks. Main limitations including lack of ablation study on design choices (i.e., sample steering alone, residual refinement alone, etc.) and lack of real-world experiments.

**Strengths:**

1. The idea is simple yet efficient, quite easy to understand and follow
2. The figures (esp. Fig. 1 & 2) are really helpful
3. Abundant simulation experiments are conducted with various baseline methods

**Weaknesses:**

1. No real-world experiments, I'm curious about the impact of learned residual actions on sim-to-real transfer
2. It appears to me that the proposed method is a simple combination of two basic ideas, how does each component contribute to the final performance improvements? I didn't see ablation studies on each design choices
3. The policy’s performance shows sensitivity to hyperparameter settings, indicating that task-specific tuning is required and potentially limiting the method’s generalizability

**Questions:**

1. Can authors elaborate more on the conclusion of "the noise action steers sampled trajectories toward the most promising mode, while the residual action enables further refinement beyond the support of the base policy" from figure 7?
2. Why DPPO is omitted on ManiSkill?
3. How does equation (7) differ from policy distillation or value decomposition in practice?

---

> ### Author Response · Authors · 2025-12-03
>
> **Q1: "No real-world experiments, I'm curious about the impact of learned residual actions on sim-to-real transfer"**
>
> **A1:** We thank the reviewer for this valuable suggestion. We have added real-world experiments on the Agibot G1 platform (see Section 5.5). Crucially, to demonstrate scalability, we applied USR to fine-tune a large-scale pre-trained Vision-Language-Action model on a "Lid Opening" task.
> As shown in Table 1, while the Base VLA model (40% success) and DSRL (75% success) struggled with precise object interaction, USR achieved a 90% success rate. This validates that USR is not limited to simulation but is capable of efficiently improving behavioral foundation models in the physical world, confirming its practical applicability.
>
> **Q2: "It appears to me that the proposed method is a simple combination of two basic ideas, how does each component contribute to the final performance improvements? I didn't see ablation studies on each design choices" "Can authors elaborate more on the conclusion of "the noise action steers sampled trajectories toward the most promising mode, while the residual action enables further refinement beyond the support of the base policy" from figure 7?"**
>
> **A2:** We are happy to clarify the contributions of the steering and residual components, as they are closely tied to the core motivation of our method. As illustrated in Figure 1, when the optimal actions lie outside the base diffusion policy’s support, which is common unless demonstrations fully cover all scenarios, steering alone (as in DSRL) can only move the policy toward a nearby mode within the base distribution, but lacks the ability to refine actions beyond that distribution. On the other hand, residual-only refinement (as in Policy Decorator) suffers from the step-size dilemma: large residual actions lead to unstable or inefficient exploration, while small residuals cannot cross gaps between modes. Moreover, residual-only methods must handle the full diversity of the base policy’s multimodal action outputs, making learning inefficient.
>
> Our approach, USR, is explicitly designed to resolve these complementary limitations by unifying steering and refinement within a single framework. The steering (noise) component guides trajectories toward promising modes of the base policy, while the residual component provides the flexibility to adapt beyond the base policy’s support when necessary. This combination yields a stable RL optimization process that retains the strengths of both mode selection and fine-grained action correction. As shown in Figure 7, the two components collaborate effectively: noise actions select the appropriate mode, and residual actions refine within or slightly outside that mode to approach the optimal solution.
> **The ablations of each design choice naturally correspond to the DSRL (steering-only) and Policy Decorator (residual-only) baselines, both of which we include across all experiments.**
>
> **Q3: "The policy’s performance shows sensitivity to hyperparameter settings, indicating that task-specific tuning is required and potentially limiting the method’s generalizability"**
>
> **A3:** As shown in Section 5.3 (Hyperparameter Study) and Figure 6, our method USR exhibits very low sensitivity to its main hyperparameters. The Noise Magnitude and Combined Critic Gradient Steps produce nearly identical performance across all tested values, and we use the same settings for these hyperparameters across almost all tasks. Both the Residual Action Scale and the Progressive Exploration Schedule also demonstrate broad workable ranges while achieving comparable sample efficiency. For example, on PlaceTwoCubeStrict, the residual-action scale is effective across 0.05–0.2 and the exploration schedule across 500K–1M steps. On OpenHingeBoxPlaceCube, the workable ranges are similarly wide (0.05–0.2 residual scale and 100K–1M exploration schedule). Overall, the hyperparameter study highlights an important strength of USR: its robustness and insensitivity to hyperparameter choices.
>
> **Q4: "Why DPPO is omitted on ManiSkill?"**
>
> **A4:** We have added DPPO baseline in ManiSkill experiments, as shown in Figure 5.

---

> > ### Author Response · Authors · 2025-12-03
> >
> > **Q5: "How does equation (7) differ from policy distillation or value decomposition in practice?"**
> >
> > **A5:** We believe that our dual-critic structure (primarily described in Equation 7) is fundamentally different from both policy distillation and value decomposition. Policy distillation, as commonly defined, transfers knowledge from a teacher policy to a student policy. Although one might draw a loose analogy in terms of high-level motivation, our approach does not distill a policy at all—instead, it distills the critic function, transferring information from the environment critic (which can be viewed as a teacher critic) into the combined critic (the student critic) to improve training stability and efficiency, as explained in Section 4.2. Value decomposition, on the other hand, is a technique from multi-agent RL in which a team-level value function is decomposed into per-agent value functions to enable decentralized policies under a centralized reward signal. This process does not involve distillation and addresses an entirely different problem setting. For these reasons, we believe our dual-critic structure is not related to value decomposition.

---

### Official Review · Reviewer_WV2j · 2025-11-02

**Soundness:** 3
**Presentation:** 3
**Contribution:** 2
**Rating:** 4
**Confidence:** 4

**Summary:**

This work, USR, augments a frozen diffusion policy with a lightweight RL actor that performs two tasks simultaneously:
1. Latent Steering: The actor outputs latent noise vectors that are fed as the diffusion model’s initial noise. This biases sampling toward promising modes in the base policy’s multimodal distribution, improving behavior selection.
2. Residual Refinement: The actor outputs a residual correction applied to the diffusion output. A scale factor $\alpha$ controls the strength of refinement, enabling fine-grained adaptation beyond the base policy’s support.

The work also releases a dataset, MultiModalBench. Results are shown in sim, on their own dataset as well as Adroit and some tasks from ManiSkill.

**Strengths:**

Paper is well motivated, with a simple method proposed to solve the task.

**Weaknesses:**

1. Lack of Real-World Validation. All experiements are in sim.
1. Unclear Critic Training Distribution: The description of the combined critic (which distills from the environment critic) omits details about the sampling distribution used for random latent-residual actions pairs.
1. Potential Policy-Critic Mismatch: The progressive gating of the application of the residual (via hyperparam $\epsilon$) during rollouts in a form of curricular learning, but it introduces a distributional gap between training and inference for the critic. The paper doesn’t analyze or justify this mismatch.
1. Less Human-generated datasets / Evaluation Design:
    * MultiModalBench rewards selecting the correct demonstration mode, a setup that inherently favors a method centered on mode steering. This could inflate the comparative advantage of USR relative to general-purpose baselines.
    * MultiModalBench, the main benchmark introduced, uses multimodal trajectories that are synthetically generated, primarily via motion planning. By virtue of using classical TAMP techniques, each “mode” is a low-variance Gaussian cluster (low Kolmogorov complexity), not a truly multimodal human-behavior distribution. Under such conditions, latent steering can easily select the correct cluster, and residual refinement only needs to make small local adjustments.
    * The paper’s only human-generated dataset results, Adroit, shows no clear advantage for USR over DSRL, the very baseline it aims to supersede. This pattern (visible in Figure 5) suggests that USR’s benefit may not generalize beyond synthetic multimodality.
    * To address this, the authors should evaluate on RoboMimic (as DSRL did). This helps test robustness on naturally diverse human demonstrations where intra-mode variance and cross-mode overlap are much higher.
1. Residual actor expressivity question: For USR to succeed on truly multimodal human data, the residual actor must itself represent multiple behavior modes. However, the paper omits implementation details. If it is a simple MLP producing a unimodal residual distribution, it cannot express multimodal refinements, which would limit generalization in richer human-collected settings.

**Questions:**

1. Considering the use of additional (2 critics and a unified actor) models, What are the training times and computational needs of the method?
1. What are the implementation details of the lightweight actor and the 2 critics? Some details, such as architecture, are missing in appendix D.
1. What is the implication of having a UTD ratio <1 ? With UTD 0.25, does that mean you discard $\frac{3}{4}$ of the training data? Please explain this further.

---

> ### Author Response · Authors · 2025-12-03
>
> **Q1: "Lack of Real-World Validation. All experiments are in sim."**
>
> **A1:** We thank the reviewer for this valuable suggestion. We have added real-world experiments on the Agibot G1 platform (see Section 5.5). Crucially, to demonstrate scalability, we applied USR to fine-tune a large-scale pre-trained Vision-Language-Action model on a "Lid Opening" task.
>
> As shown in Table 1, while the Base VLA model (40% success) and DSRL (75% success) struggled with precise object interaction, USR achieved a 90% success rate. This validates that USR is not limited to simulation but is capable of efficiently improving behavioral foundation models in the physical world, confirming its practical applicability.
>
>
> **Q2: "The description of the combined critic (which distills from the environment critic) omits details about the sampling distribution used for random latent-residual actions pairs."**
>
> **A2:** We apologize for the omission. In our implementation, the random combined actions for distillation are sampled as follows:
>
> - Latent noise: sampled from a standard Gaussian distribution, matching the prior of the base diffusion policy model.
>
> - Residual action: sampled from a Uniform distribution over the action space range. To match the progressive exploration strategy used during rollouts, we also apply a random mask to residual actions during distillation. With probability $(1 - \epsilon)$, we set $a^{res} = 0$.
>
> **Q3: "The progressive gating of the application of the residual (via hyperparam ) during rollouts in a form of curricular learning, but it introduces a distributional gap between training and inference for the critic."**
>
> **A3:**
> There is no distributional mismatch on the critic side. The environment critic takes the final executed actions in the environment as input, so the progressive exploration schedule does not affect it. When the residual action is not introduced, the residual component provided to the combined critic is simply a zero vector, which introduces neither ambiguity nor distributional shift.
>
> **Q4: "Less Human-generated datasets / Evaluation Design"**
>
> **A4:** To fully address the concern regarding evaluation on human-generated datasets, we have added experiments on RoboMimic (can and square tasks), a standard benchmark containing diverse human demonstrations.
>
> To stress-test the method's ability to improve policies online, we intentionally trained a weak base policy to start from. As shown in Figure 11, DSRL failed to yield meaningful improvement on the can task and was highly unstable on square. In contrast, USR consistently reached near 100% success, proving that our residual component effectively bridges the precision gap by enabling refinement beyond the base policy's support.
>
> **Q5: "For USR to succeed on truly multimodal human data, the residual actor must itself represent multiple behavior modes. However, the paper omits implementation details. If it is a simple MLP producing a unimodal residual distribution, it cannot express multimodal refinements, which would limit generalization in richer human-collected settings."**
>
> **A5:** We clarify that the residual actor is implemented as an MLP parameterizing a unimodal Gaussian distribution. This design leverages the interplay between our two components: **latent steering acts as a mode selector that collapses the base policy's multimodal distribution into a single promising mode, effectively creating a unimodal distribution for the residual to act upon.** Consequently, the residual actor only needs to perform a local shift of this selected mode toward high-reward regions, making a unimodal correction sufficient even for complex multimodal base policies. This mechanism is empirically validated in Section 5.4, where we visualize how the steered actions form a clear single-modal distribution, which is subsequently shifted by the residual refinement to achieve task success.
>
> **Q6: "Considering the use of additional (2 critics and a unified actor) models, what are the training times and computational needs of the method?"**
>
> **A6:** USR is computationally efficient and can be easily trained on a single consumer-grade GPU. On a single NVIDIA RTX 4090, training for 1 million environment steps in our MultiModalBench takes approximately 6 to 7 hours. The computational cost of USR is almost identical to DSRL. Since the unified actor and critics are lightweight MLPs, the only architectural difference is the increase in the actor's output layer and critics' input layer (to include residuals). The primary computational bottleneck remains the iterative denoising process of the base diffusion policy model, which is shared by both methods.

---

> > ### Author Response · Authors · 2025-12-03
> >
> > **Q7: "What are the implementation details of the lightweight actor and the 2 critics? Some details, such as architecture, are missing in appendix D."**
> >
> > **A7:** We have included the architectural details of the lightweight actor and critic in Appendix D.2.3.
> >
> > **Q8: "What is the implication of having a UTD ratio <1 ? With UTD 0.25, does that mean you discard  of the training data? Please explain this further."**
> >
> > **A8:** A UTD ratio of 0.25 means we perform 1 gradient update for every 4 environment steps collected. This does not imply discarding data. With a batch size of 1024, each gradient update samples 1024 transitions from the replay buffer, covering significantly more samples than the 4 newly collected ones. All collected transitions are stored in the buffer for repeated reuse.
> >
> > We also conducted a hyperparameter study on the UTD. As shown in Figure 16, setting UTD = 1.0 yields only marginal gains in sample efficiency while significantly slowing down training. Consequently, we select a UTD value that best trades off sample efficiency against training speed. Check Appendix E.7 for more detailed information.

---

### Official Review · Reviewer_H4pE · 2025-11-05

**Soundness:** 2
**Presentation:** 3
**Contribution:** 2
**Rating:** 4
**Confidence:** 4

**Summary:**

The paper proposes USR, a method that uses RL to steer and improve a pre-trained frozen diffusion policy. USR provides a unified framework that combines concepts from latent-noise steering and residual-RL approaches. Specifically, it trains an RL actor to output the concatenation of an initial noise term and a residual action. The initial noise can guide the policy toward promising modes, and the residual action refines the output to extend performance beyond the support of the pre-trained policy. Experiments on simulation manipulation benchmarks, including a newly proposed MultiModalBench, show that USR can outperform both DSRL and Residual RL.

**Strengths:**

The paper is clearly written overall, and the idea of unifying initial-noise steering with residual RL is simple yet well motivated. The practical algorithm is well-designed and combines dual critics with distillation. The simulated evaluation seems promising, with the USR outperforming all the baselines.

**Weaknesses:**

The main concern is the selection of the task benchmarks. Most evaluations are performed on the newly introduced MultiModelBench, making it difficult to calibrate task difficulty and judge how prior methods perform. Notably, DSRL does better on Adroit and ManiSkill (Figure 5) but struggles on MultiModelBench, raising the concern that the benchmark may have been constructed in a way that disadvantages baselines while favoring the proposed approach. I recommend adding additional benchmarks such as RoboMimic, which is widely used in prior work (e.g., DPPO and DSRL). Please see the next section for further questions.

**Questions:**

- I’m a bit confused about the design of MultiModalBench. The authors mention that the benchmark is designed to “evaluate the ability of adjusting multimodal policies toward the desired single-mode behavior.” Then why does DSRL struggle to improve? Isn’t this a mode-selection problem, since the dataset already contains all modes?

- The results note that DSRL is "prone to collapse during training". Given that DSRL is conceptually similar and USR shares similar hyperparameters, why is DSRL unstable while USR is stable?

- While both USR and DSRL use the dual-critic version, have the authors tried naively training a single critic in the latent space (similar to the DSRL-SAC variant mentioned in their paper)?

- Related to the question above, the motivation for noise-aliased DSRL is that it can leverage offline demonstrations to further improve sample efficiency. Have the authors tried combining offline demonstrations with USR to see if it provides additional benefits? Including this ablation would make the analysis more complete and interesting.

- The implementation details of the unified actor and critic are missing. What is the architecture?

- Why is a discount factor (gamma) of 0.97 used across the experiments? Have the authors performed a sweep over this value? It seems quite low compared to the values typically used in prior work (0.99 or 0.999).

I’d be happy to reconsider my score once these points are addressed.

---

> ### Author Response · Authors · 2025-12-03
>
> **Q1: "I recommend adding additional benchmarks such as RoboMimic, which is widely used in prior work (e.g., DPPO and DSRL)."**
>
> **A1:** Thanks for your suggestion. In the revision, we have added experiments on two representative RoboMimic tasks: can and square. We focused on these two tasks as lift is relatively trivial for base policies, and transport requires excessive computational resources given the rebuttal timeline.
>
> To stress-test the method's ability to improve policies online, we intentionally trained a weak base policy. This creates a challenging setting where the pre-trained distribution does not consistently cover successful task executions, forcing the adaptation method to explore beyond the initial support.
>
> As shown in the newly added Figure 11 in the Appendix, the results strongly highlight the limitations of DSRL and the necessity of USR. On the can task, DSRL, which performs pure noise steering, failed to yield any meaningful improvement. Results on the square task exhibit high variance, with very few seeds succeeding in improving upon the base policy. We suspect this failure stems from the inherent constraints of the steering approach. While the base policy's pre-trained distribution likely captures the correct behavioral mode, the actions within this mode lack the necessary precision to obtain the sparse reward reliably.
>
> In contrast, USR consistently improved the success rate to near 100% on both tasks, demonstrating robust stability. This performance gap highlights the need for a residual component, which allows the policy to refine actions and explore effectively outside the base policy's support. By enabling out-of-distribution corrections, USR can bridge the gap between an imperfect pre-trained policy and the precision required for task success.
>
> **Q2: "The authors mention that the benchmark is designed to 'evaluate the ability of adjusting multimodal policies toward the desired single-mode behavior.' Then why does DSRL struggle to improve? Isn’t this a mode-selection problem, since the dataset already contains all modes?"**
>
> **A2:** We acknowledge that our original statement was imprecise. In the revision, we have updated the description in Section 5.1.1 to explicitly state that the benchmark evaluates **both** efficient mode selection **and** fine-grained action refinement.
> MultiModalBench tasks are not purely mode-selection problems. As shown in Figure 4, the success rate of the Base Policy (dashed black line, where **any** mode is considered successful) is not perfect (often well below 100%). This performance roughly represents what pure mode selection can achieve. Even if DSRL steers to the correct mode, it is still limited by inherent flaws or the lack of precision in the base policy's outputs. The agent must not only select the promising mode but also fine-tune the actions to surpass the base policy's capabilities. DSRL fails to achieve high success rates because it is more constrained by the base policy's support. USR performs better because its residual component enables it to explore beyond the base policy's pretrain distributions.
>
> **Q3: "The results note that DSRL is 'prone to collapse during training'. Given that DSRL is conceptually similar and USR shares similar hyperparameters, why is DSRL unstable while USR is stable?"**
>
> **A3:** The instability arises from a fundamental difference in the underlying optimization landscape.
>
> - **Complex vs. Direct Mapping.** In DSRL, the control variable (noise) affects the action only through the diffusion model, which is a complex and opaque transformation. Achieving fine-grained adjustments, therefore, requires broad exploration within this highly nonlinear noise space, where gradients fluctuate sharply, and the risk of getting trapped in local optima is high.
> - **Predictable Control.** In contrast, USR introduces a residual branch that adjusts the final action through simple addition. This gives the actor a clear and predictable way to influence the output.
> - **Decoupled Roles.** USR also separates responsibilities. Noise is used for coarse mode selection, while the residual term provides precise refinement. This structure allows the lightweight actor to achieve stable improvements via the residual path and prevents the collapse that often occurs when the noise pathway is forced to handle both coarse and fine control simultaneously.

---

> > ### Author Response · Authors · 2025-12-03
> >
> > **Q4: "While both USR and DSRL use the dual-critic version, have the authors tried naively training a single critic in the latent space (similar to the DSRL-SAC variant mentioned in their paper)?"**
> >
> > **A4:** Following the reviewers’ request, we conducted additional experiments using the SAC variant introduced in the original DSRL paper, applied to both USR and DSRL. The results in Figure 12 show that DSRL-SAC and USR-SAC consistently underperform their noise-aliasing counterparts, our original method. This finding aligns with the observations reported in the DSRL paper, which highlights that the noise-aliasing variant reduces unnecessary exploration in the latent-noise space and thereby improves sample efficiency. Check Appendix E.3 for more detailed information.
> >
> > **Q5: "Have the authors tried combining offline demonstrations with USR to see if it provides additional benefits?"**
> >
> > **A5:** In response to the reviewer’s request, we incorporated offline demonstrations into USR, and the experimental results in Figure 13 show that this does not provide a noticeable improvement in sample efficiency. Check Appendix E.4 for more detailed information.
> >
> > **Q6: "The implementation details of the unified actor and critic are missing."**
> >
> > **A6:** We have included the architectural details of the unified actor and critic in Appendix D.2.3.
> >
> > **Q7: "Why is a discount factor (gamma) of 0.97 used across the experiments? Have the authors performed a sweep over this value?"**
> >
> > **A7:** We follow several principles when choosing the discount factor. First, we use a consistent value within each benchmark to avoid excessive tuning. Second, we adopt discount factors used in prior work whenever possible to ensure fair comparisons with existing baselines. Third, all methods in our experiments use the same discount factor to maintain fairness. In response to the reviewer’s request, we also performed a sweep of discount factors on MultiModalBench and Adroit. The results show that a value of 0.97 performs best on MultiModalBench, while on Adroit, all tested values achieve similar performance. Check Appendix E.6 for more detailed information.

---

### Author Response · Authors · 2025-12-03
**General Response by Authors**

Dear Area Chairs and Senior Area Chairs,

We would like to briefly summarize the main concerns from reviewers and how we properly resolve them in the rebuttal to aid your decision.

---

**Real-World Experiments (Reviewer WV2j, 2Zek, Ke1h)**

We have added real-world experiments on the Agibot G1 platform (see Section 5.5). Crucially, to demonstrate scalability, we applied USR to fine-tune a large-scale pre-trained Vision-Language-Action model on a "Lid Opening" task. As shown in Table 1, while the Base VLA model (40% success) and DSRL (75% success) struggled with precise object interaction, USR achieved a 90% success rate. This validates that USR is not limited to simulation but is capable of efficiently improving behavioral foundation models in the physical world, confirming its practical applicability. We have included the real robot rollout videos in supplementary materials.

**RoboMimic/Additional Benchmarks/more human generated datasets (Reviewer H4pE, WV2j)**

In response to the request from reviewers to add RoboMimic and experiment with more human generated datasets, we have added experiments on two representative RoboMimic tasks: can and square. USR consistently outperforms DSRL, improving the success rate to near 100% on both tasks. Check Appendix E.2 for more detailed information.

**Ablation Study on Dual-Critic vs. Single-Critic Architectures (Reviewer H4pE)**

Following the reviewers’ request, we conducted additional experiments using the SAC variant introduced in the original DSRL paper, applied to both USR and DSRL. The results show that DSRL-SAC and USR-SAC consistently underperform their noise-aliasing counterparts—our original method. This finding aligns with the observations reported in the DSRL paper, which highlights that the noise-aliasing variant reduces unnecessary exploration in the latent-noise space and thereby improves sample efficiency. Check Appendix E.3 for more detailed information.

**USR with offline demonstrations (Reviewer H4pE)**

In response to the reviewer’s request, we incorporated offline demonstrations into USR, and the experimental results show that this does not provide a noticeable improvement in sample efficiency. Check Appendix E.4 for more detailed information.

**Hyperparameter Study on Discount Factor (Reviewer H4pE)**

We follow several principles when choosing the discount factor. First, we use a consistent value within each benchmark to avoid excessive tuning. Second, we adopt discount factors used in prior work whenever possible to ensure fair comparisons with existing baselines. Third, all methods in our experiments use the same discount factor to maintain fairness. In response to the reviewer’s request, we also performed a sweep of discount factors on MultiModalBench and Adroit. The results show that a value of 0.97 performs best on MultiModalBench, while on Adroit, all tested values achieve similar performance. Check Appendix E.6 for more detailed information.

**Hyperparameter Study on Update-to-Data Ratio (Reviewer WV2j)**

We conducted a hyperparameter study on the UTD. As shown in Figure 16, setting UTD = 1.0 yields only marginal gains in sample efficiency while significantly slowing down training. Consequently, we select a UTD value that best trades off sample efficiency against training speed. Check Appendix E.7 for more detailed information.

**Imbalanced Mode Dataset (Reviewer Ke1h)**

The dataset used in our main MultiModalBench experiments, as shown in Figure 4, contains an
equal number of demonstrations for each behavior mode. To more comprehensively evaluate our
method under diverse data distributions, we additionally conduct experiments in two scenarios where
the expected mode is underrepresented in the dataset.
As shown in Figure 14, our method maintains robust performance in different levels of underrepresented
(10% and 30%) settings, whereas DSRL experiences substantial degradation when
the expected mode is underrepresented. Check Appendix E.5 for more detailed information.

**Other Conceptual Problems (Reviewer H4pE, WV2j, 2Zek, Ke1h)**

We provide detailed response to conceptual questions proposed by all four reviewers.

---

We believe we have thoroughly addressed all concerns raised by the four reviewers during the rebuttal period. We hope the Area Chairs and Senior Area Chairs will take this into consideration and reach a well-informed decision.

---

### Meta-Review · Area_Chair_X8rg · 2026-01-07

**Summary:**

Reviewers appreciated the clear motivation of the USR framework, which unifies latent steering for mode selection with residual refinement for out-of-distribution adaptation. However, the submission initially faced significant skepticism regarding its empirical breadth and the generalizability of its findings:
- Reviewers `WV2j` `2Zek` `Ke1h` critiqued the total absence of real-world experiments in the initial submission, which made it difficult to assess the method's practical applicability to physical robots.
- Reviewers `H4pE` and `WV2j` expressed concern that the newly introduced MultiModalBench relied on synthetically generated, low-variance motion planning trajectories. They worried this favored latent steering and might not represent the complexity of true human-behavior distributions.
- Reviewer `H4pE` noted that the baseline DSRL performed competitively on established benchmarks like Adroit and ManiSkill, suggesting the proposed method’s advantages might be benchmark-specific.
- Reviewers `H4pE` `WV2j` recommended evaluating on RoboMimic to test robustness against high-variance human demonstrations.

Prior to the rebuttal, the scores were consistently below the acceptance threshold, primarily at Marginally Below (4). In the rebuttal, the authors made a substantial effort to address these gaps:
- They added a real-robot "Lid Opening" experiment on the Agibot G1 platform, by fine-tuning a Vision-Language-Action (VLA) model.
- Results were added for two RoboMimic tasks (can and square), where USR outperformed DSRL.
- The authors provided extensive ablations on critic architectures, imbalanced mode datasets, and hyperparameter sensitivity.

Despite these additions, the AC concluded that the empirical story remains overly reliant on the authors' own benchmark. The added evidence, while promising, was deemed insufficient in scope to demonstrate the robustness and generality required for the broad claims made in the paper. Consequently, the final recommendation is Reject.

**Reviewer Concerns:**

Across reviews, the central concern is insufficient external validation. Reviewers consistently noted the initial lack of real-world evaluation and that the empirical story relies heavily on the newly introduced MultiModalBench, making it difficult to calibrate task difficulty and assess generality or potential benchmark bias. I appreciate the authors’ effort in adding additional experiments in the rebuttal, including a real-robot lid-opening task and two RoboMimic tasks, which meaningfully strengthen the submission. However, these additions are still limited in scope relative to the paper’s broad claims about sample-efficient online improvement of diffusion-based manipulation policies. Other concerns such as ablations, stability compared to DSRL, implementation details, and hyperparameter sensitivity are largely addressed in the rebuttal, but the main issue remains, the current evidence does not yet convincingly demonstrate robustness across diverse, established simulation suites and broader real-world settings, and fully addressing this would require substantial additional experimentation and likely a restructuring of the evaluation.

**Reviewer Scores:**

Even if some reviewers might increase their ratings slightly due to the added rebuttal experiments, the shared concern about missing/insufficient real-world validation and the over-reliance on MultiModalBench suggests the submission remains below the bar for acceptance in its current form. I encourage the authors to strengthen a future submission by adding more extensive real-world evaluations (multiple tasks, objects, and conditions; longer-horizon multi-stage behaviors; and stronger evidence of sim-to-real robustness), and by expanding simulation results to additional established benchmarks beyond MultiModalBench (e.g., broader RoboMimic coverage, and other widely used suites such as ManiSkill and Adroit variants with clear baselines and protocols). It would also help to include clearer cross-benchmark comparisons and analysis that demonstrate when steering, residual refinement, and their unified training are necessary, and where the method may fail.

---

### Decision · Program_Chairs · 2026-01-26

Reject